# An Experimental Study of Grouping Mutation Operators for the Unrelated Parallel-Machine Scheduling Problem

Octavio Ramos-Figueroa * , Marcela Quiroz-Castellanos , Efrén Mezura-Montes and Nicandro Cruz-Ramírez

Artificial Intelligence Research Institute, Universidad Veracruzana, Campus Sur, Calle Paseo Lote II, Sección Segunda 112, Nuevo Xalapa, Veracruz 91097, Mexico
* Correspondence: oivatco.rafo@gmail.com

**Abstract:** The Grouping Genetic Algorithm (GGA) is an extension to the standard Genetic Algorithm that uses a group-based representation scheme and variation operators that work at the group-level. This metaheuristic is one of the most used to solve combinatorial optimization grouping problems. Its optimization process consists of different components, although the crossover and mutation operators are the most recurrent. This article aims to highlight the impact that a well-designed operator can have on the final performance of a GGA. We present a comparative experimental study of different mutation operators for a GGA designed to solve the Parallel-Machine scheduling problem with unrelated machines and makespan minimization, which comprises scheduling a collection of jobs in a set of machines. The proposed approach is focused on identifying the strategies involved in the mutation operations and adapting them to the characteristics of the studied problem. As a result of this experimental study, knowledge of the problem-domain was gained and used to design a new mutation operator called 2-Items Reinsertion. Experimental results indicate that the state-of-the-art GGA performance considerably improves by replacing the original mutation operator with the new one, achieving better results, with an improvement rate of 52%.

**Keywords:** grouping genetic algorithm; grouping mutation operator; grouping problem; unrelated parallel-machine scheduling



## 1. Introduction

Over the last decades, the interest of the scientific community in solving Combinatorial Optimization Problems (COPs) has grown considerably since these types of problems emerge in many practical issues in industry, logistics, and engineering. In general, the optimization of a COP comprises the search of the suitable values for a set of discrete variables, so that the objective function is optimized, satisfying the given conditions and constraints. Thus, the solution of this type of problems can involve a feasible disposition, grouping, order, or selection of discrete objects that typically are finite in number [1]. It is well-known that many COPs have high complexity, and in the worst-case scenario, there is no efficient algorithm that solves all their possible cases optimally. Such problems belong to the NP-hard class [2]. In this order of ideas, this work focuses on grouping problems, a special type of COPs that in general consist of looking for an efficient arrangement of a set of elements among a collection of groups [1].

Parallel-Machine Scheduling (PMS) is a classical NP-hard grouping problem, consisting of looking for the most efficient sequential scheduling of a set of $n$ jobs $N = \{j_1, \ldots, j_n\}$ among a collection of $m$ parallel-machines $M = \{i_1, \ldots, i_m\}$, in such a way that each machine $i$ can process only one job $j$ at a time, and each job $j$ must be processed by a single machine $i$ [3].

The PMS variants can consider different parameters in the problem definition, such as resource and scheduling environments, job characteristics, and optimization criteria, among others. The most general classification of PMS problems is according to the machine

environment. In this sense, this work focuses on a variant that belongs to the class with unrelated machines, i.e., each machine can require a different time to process each job, and there is not a behavior pattern with respect to the speed of the machines with a machine always being the fastest or the slowest one (Unrelated Parallel-Machine Scheduling, UPMS). This problem family has received much recognition due to its numerous real-world applications [4–6]. Although a large number of mathematical models have been proposed, the exact approaches can solve only small instances in a reasonable time [7]. Given the complexity of several UPMS variants, most approaches are metaheuristic algorithms, such as local searches, swarm intelligence, and evolutionary algorithms. The state of the art contains local searches such as the Hill Climbing [8], the Iterated Greedy Algorithm [9], the Variable Neighborhood Descent [10], and the GRASP Algorithm [11]. In the same spirit, the literature includes several swarm intelligence algorithms, such as the Worm Optimization Algorithm [12], the Firefly Algorithm [13], the Artificial Bee Colony [14], and the Fruit Fly Optimization Algorithm [15]. Additionally, we identified several evolutionary algorithms such as the Genetic Algorithm [16], the Genetic Programming [17], and the Imperialist Competitive Algorithm with memory [18]. Finally, the specialized literature includes some memetic algorithms [19,20]. The literature review reveals that there are a wide variety of UPMS problems, each with particular characteristics and challenges. Given the increasing appearance of these problems, there exists a trend to explore the algorithmic behavior of different metaheuristic approaches that can work well or badly according to the properties of the variant of the problem to solve. One of the main challenges in the development of high-performance algorithms for UPMS problems is the design of efficient strategies that work together with the features of the problem variant to find high-quality solutions.

This work addresses the UPMS variant known as the $R||C_{max}$ problem, where the machines $\{i_1, \ldots, i_m\}$ are unrelated, jobs $\{j_1, \ldots, j_n\}$ have no-preemptions, and the objective of interest is the reduction of the maximum completion time $C_{max}$, i.e., the processing time $C_i$ required by the machine $i$ that finishes at the end.

It is well-known that the problem $R||C_{max}$ belongs to the class NP-hard [2]. Hence, over the past forty years, different approaches have been studied to try to solve it efficiently. The specialized literature includes deterministic methods [21,22], two-phase algorithms (or rounding methods) [23,24], and branch and bound algorithms [3,25]. The literature also includes distinct metaheuristic algorithms for $R||C_{max}$, covering proposals based on local searches [3,26], the swarm intelligence algorithm Particle Swarm Optimization (PSO) [1], the Genetic Algorithm (GA) [27], and some hybrid approaches [3]. According to the scope of this review, the approaches based on local searches have shown the best performance on solving the problem $R||C_{max}$. The state of the art highlights the results reached by the Iterated Greedy Local Search (NVST-IG+) proposed by Fanjul-Peyro and Ruiz in 2009, considered one of the best solution methods designed for the problem of interest so far. The success key of the NVST-IG+ performance is the incorporation of some techniques to control the way in which the jobs and machines are selected and manipulated during the construction of the neighborhoods [26].

In [1], we presented one of the most recent related works; the experimental results suggested that a GA with a group-based representation GGA has a better performance than a GA with an extended permutation solution encoding and a PSO with a machine-based representation scheme for the 1400 test instances studied. Such GGA was an adaptation of the GGA-CGT designed by Quiroz-Castellanos et al. for the Bin Packing Problem [28]. According to Quiroz-Castellanos et al., the performance of the GGA-CGT is related mainly to the mutation operator, which alone is capable of finding quality solutions. The mutation is one of the most used genetic operators in GGAs. Commonly, mutation operators promote the exploration of the search space by slightly altering the solution genetic material. This behavior is useful for a GGA mainly when it is converging to a local optimum since it provides the capacity to redirect the search to other areas. Section 2.5 includes an experimental study with different parameter configurations that allows observing how the performance of the GGA proposed in [1] is mainly related to the crossover operator,

while the mutation operator has a low impact. The above motivates this work that aims to study the performance of different grouping mutation operators to identify the strategies that they use and that positively impact their performance, to employ them in the design of a new operator, and to incorporate that operator into the GGA in order to improve its performance when solving $R||C_{max}$.

This paper continues as follows. Section 2 describes the components and the problem-domain heuristics of the GGA for $R||C_{max}$. Section 3 reviews the state-of-the-art grouping mutation operators. Section 4 contains the experimental design proposed to analyze the impact of different strategies in the performance of grouping mutation operators. Section 5 compares the GGA performance with the new and the old mutation operators to analyze the improvement rate. Finally, Section 6 summarizes the conclusions and future paths of research.

## 2. Grouping Genetic Algorithm for $R||C_{max}$

The state of the art suggests that the GGA is one of the most used metaheuristics to solve grouping problems. Such popularity is related to its promising results and its flexibility to adopt new ideas to handle the constraints and conditions of the problem to be solved [1,29,30].

The GGA is an extension to the standard GA; therefore, it has a similar procedure. The GGA starts with the generation of the initial population, generally in a random way. Next, selection strategies and variation operators, mainly crossover and mutation, are used iteratively so as to find better solutions. Each iteration represents a generation that starts utilizing a selection strategy to pick some individuals of the population based on their fitness values; then, the genetic material of the selected individuals is recombined with the crossover operator to generate offspring. Subsequently, the offspring are added to the population using a replacement strategy. Finally, some individuals, chosen with a selection strategy, are slightly modified with the mutation operator. In this way, the GGA iterates performing the before-mentioned procedure until some stopping criterion (e.g., the maximum number of generations, the maximum search time, convergence of solutions, or finding an optimal solution) is met.

One of the main features of the GGA is the group-based scheme that it uses to encode and manage solutions in the search space. According to Falkenauer, this is a more natural way of representing solutions to grouping problems. Moreover, it helps to reduce the search space since it produces fewer isomorphic solutions than a traditional representation scheme [31]. In this encoding, each gene represents a group that contains the collection of elements that correspond to it. Therefore, the length of a solution is equal to the number of groups that it includes.

Another important aspect to consider when developing a GGA is the design of variation operators such as crossover and mutation since they must work at the group level. With this feature, operators can perform procedures in a more controlled way, determining which groups and elements vary according to the constraints and objectives of the problem to solve. The crossover operator uses two or more solutions of the current population to recombine their genetic material, creating offspring with new characteristics. This operator is used to give GGA the ability to converge on the most promising areas identified during the search. One of the advantages of crossover operators for the group-based encoding is that they can use the quality of the groups to determine how parents transmit the genetic material to their offspring to perform a more controlled search. On the other hand, the mutation operator provides GGA the ability to explore new areas of the search space, producing small modifications to the genetic material of some solutions. This procedure is helpful for a GGA, mainly to address highly constrained grouping problems, where there are large possibilities of converging to local optimums. These slight alterations performed by the mutation operator can generate solutions in other regions of the search space, avoiding premature convergence [32].

The next sections describe the elements of the state-of-the-art GGA for $R||C_{max}$, the object of study in this work, including the population initialization strategy, the variation operators, selection and replacement strategies, and the problem-domain heuristics. This algorithm is an adaptation of the Grouping Genetic Algorithm with Controlled Gene Transmission (GGA-CGT) introduced by Quiroz-Castellanos et al. to solve the Bin Packing problem [28]. The details of the heuristic used to generate the initial population, as well as the mutation and crossover operators appear in the work of Ramos-Figueroa et al. [1], while the remaining mechanisms and operators, as well as the parameter settings can be consulted in the work of Quiroz-Castellanos et al. [28].

### 2.1. Genetic Encoding, Fitness Function, and Initial Population

The GGA uses the group-based representation scheme to encode and manipulate solutions, where each machine $i$ is a gene (or group) $G_i$ that will include a set of jobs. Therefore, all solutions have the same number of genes, equal to the number of machines $m$. The quality of each machine $i$ is equal to the time it takes to process its assigned jobs, denoted as $C_i$. Thus, the quality of a solution $C_{max}$ is equal to the $C_i$ value of the machine with the longest processing time. The initial population is generated in a random manner by running the well-known Min() heuristic on random permutations of the $n$ jobs [33]. For each job $j$, Min() calculates the equation $C_i = C_i + p_{ij}$ for all the machines, where $p_{ij}$ indicates the time that the machine $i$ needs to process the job $j$. In this way, Min() assigns the job $j$ to the machine $i$ that generates the lowest $C_i$ value.

Figure 1 describes the procedure followed by the population initialization strategy. To give a comprehensive description, Figure 1a includes an example instance $I$ represented as a matrix with $m = 4$ machines depicted by the columns and $n = 10$ jobs represented by the rows. Thus, the example starts from a permutation (Figure 1b) of the ten jobs, $\{j_9, j_5, j_2, j_6, j_3, j_8, j_4, j_7, j_1, j_{10}\}$, used to generate the partial solution, shown in Figure 1c. The construction of the partial solution can be calculated from the first nine jobs in the permutation, $\{j_9, j_5, j_2, j_6, j_3, j_8, j_4, j_7, j_1,\}$ and the instance $I$ using the heuristic Min(). To exemplify how this heuristic Min() works, Figure 1d shows a complete solution, resulting from the assignment of the last job in the permuted list (i.e., $j_{10}$) to the solution. Therefore, following the Min() procedure mentioned above, the processing time $C_i$ of each machine plus the time that they require to process the job $j_{10}$ results in the following way: $C_1 = 26 + 8$, $C_2 = 25 + 20$, $C_3 = 20 + 18$, and $C_4 = 10 + 28$. Hence, Min() assigned the job $j_{10}$ to the machine $i_1$ since it generated the lowest $C_i$ value. It is important to note that if two or more machines produce the same $C_i$ value, this allocation heuristic assigns the job in turn to the machine $i$ that appears first from $i_1$ to $i_m$. Finally, Figure 1d also shows the fitness value of the generated solution that is equal to the longest processing time $C_i$, in this case, the $C_1 = 34$, outlined in bold.

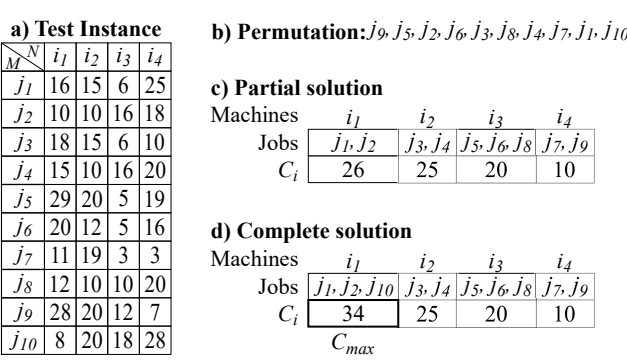

**Figure 1.** Population initialization strategy.

### 2.2. Adapted Gene-Level Crossover Operator

The GGA uses the Adapted Gene-Level Crossover (AGLX) operator, a variant of the GLX operator proposed by Quiroz-Castellanos et al. [28] that produces two children

solutions by using two parent solutions. Algorithm 1 presents the procedure of AGLX. We denote $S' = \text{Sort}(S)$ the solution derived from $S$ by sorting its machines in increasing order concerning its $C_i$ values (lines 1 and 2). Thus, AGLX first transmits the machines that process their jobs fastest and then the slowest ones (lines 3–6). In this way, the first child $C_1$ starts inheriting the fastest machine from the first parent $S_1$, next the fastest machine from the second parent $S_2$, then the second-fastest machine from the first parent $S_1$, and so on (line 4). Similarly, the second child $C_2$ receives genes alternately from both parents but starting with the fastest machine from the second parent $S_2$ (line 5). We denote $C = \text{Inherit}(C, i_a, i_b)$ the child solution $C$ upgraded with the machines $i_a$ and $i_b$, one for each parent. It is important to remark that before inheriting each machine, the Inherit() function verifies that it has not already been transmitted by the other parent to the child $C$. Otherwise, the machine is discarded. Likewise, before inheriting each job, this function validates that it has not already been transmitted. Otherwise, it is discarded to avoid infeasible solutions. It is important to note that in most of cases this procedure generates infeasible solutions, since some jobs can be missed during the crossover process. Therefore, it is necessary to re-insert the jobs to transform the solutions into feasible ones (lines 7 and 8). We denote $MJ[] = \text{MissedJobs}(C)$ the set of jobs missed during the genetic material transmission of a child $C$. Finally, the missed jobs $MJ$ are permuted and re-inserted with the Min() heuristic described above (lines 9–12). We denote $MJ[]' = \text{Permute}(MJ[])$ the set of jobs derived from $MJ[]$ by permuting it with a uniform distribution and $C' = \text{Min}(C, MJ[]')$ the child solution obtained from the re-insertion of the jobs in $MJ[]'$ to the solution $C$.

---

**Algorithm 1** AGLX operator

> **Input:** Two parent solutions $S_1$ and $S_2$, and the number of machines $m$.
> **Output:** Two offspring solutions $C_1'$ and $C_2'$.

1: $S_1' = \text{Sort}(S_1)$;
2: $S_2' = \text{Sort}(S_2)$;
3: **for each** machine $i$ in $S_1'$ and $S_2'$ **do**
4:     $C_1 = \text{Inherit}(C_1, S_1'[i], S_2'[i])$;
5:     $C_2 = \text{Inherit}(C_2, S_2'[i], S_1'[i])$;
6: **end for**
7: $MJ_1[] = \text{MissedJobs}(C_1)$;
8: $MJ_2[] = \text{MissedJobs}(C_2)$;
9: $MJ_1[]' = \text{Permute}(MJ_1[])$;
10: $MJ_2[]' = \text{Permute}(MJ_2[])$;
11: $C_1' = \text{Min}(C_1, MJ_1[]')$;
12: $C_2' = \text{Min}(C_2, MJ_2[]')$;
13: **end process.**

---

Figure 2 describes the process of the AGLX operator with an example that contains two parent solutions for the test instance of Figure 1a with four machines (groups). The ten jobs, from $j_1$ to $j_{10}$, are distributed among the four machines, from $i_1$ to $i_4$, and the time that each machine $i$ requires to process its assigned jobs from $C_1$ to $C_4$ is stored in the vector $C_i$. Figure 2a depicts the transmission process. Therefore, it shows the two parents with their groups in increasing order, which indicates the gene transmission sequence, i.e., from best (Lowest $C_i$) to worst (Highest $C_i$). Figure 2b indicates the way the repeated genetic material is handled. Thus, it contains the two solutions produced during the transmission process, which only keep the machine $i$ of the parent in which it appears first according to the gene transmission sequence. Furthermore, this figure includes the repeated jobs, highlighted in bold, that must be removed from the machine with the highest processing time $C_i$. Lastly, this figure shows a list with the jobs missed $MJ$ during the transmission process. Figure 2c contains the partial solution resulting from the transmission process without the repeating genetic material, as well as a permutation of the jobs in $MJ$. Finally, Figure 2d shows the

complete solutions resulting from the assignment of the missed jobs with the heuristic Min(). The processing time $C_i$ of each machine $i$, as well as the operations performed by the Min() heuristic to assign the missed jobs, can be calculated using the example instance presented in Figure 1a.

Given two parent solutions for the test instance of Figure 1a, the Adapted Gene-level crossover operator (AGLX) proposed by Ramos-Figueroa *et al.* [1] works as follows:

**a) Transmission process**

| Machines | $i_4$ | $i_3$ | $i_2$ | $i_1$ | |
|---|---|---|---|---|---|
| Jobs | $j_7, j_9$ | $j_5, j_6, j_8$ | $j_3, j_4$ | $j_1, j_2, j_{10}$ | First Parent |
| $C_i$ | 10 | 20 | 25 | 34 | |

| Machines | $i_2$ | $i_3$ | $i_1$ | $i_4$ | |
|---|---|---|---|---|---|
| Jobs | $j_2$ | $j_1, j_3, j_6, j_7$ | $j_5, j_9$ | $j_4, j_8, j_{10}$ | Second Parent |
| $C_i$ | 10 | 20 | 57 | 68 | |

**b) Repeated genetic material**

| Machines | $i_4$ | $i_2$ | $i_3$ | $i_1$ | | MJ |
|---|---|---|---|---|---|---|
| Jobs | $j_7, j_9$ | $j_2$ | $j_5, j_6, j_8$ | $\boldsymbol{j_5, j_9}$ | First Child | $j_1, j_3, j_4, j_{10}$ |
| $C_i$ | 10 | 10 | 20 | 57 | | |

| Machines | $i_2$ | $i_4$ | $i_3$ | $i_1$ | | MJ |
|---|---|---|---|---|---|---|
| Jobs | $j_2$ | $j_7, j_9$ | $j_1, j_3, j_6, \boldsymbol{j_7}$ | $\boldsymbol{j_5, j_9}$ | Second Child | $j_4, j_8, j_{10}$ |
| $C_i$ | 10 | 10 | 20 | 57 | | |

**c) Partial solution**

| Machines | $i_1$ | $i_2$ | $i_3$ | $i_4$ | | Permutation |
|---|---|---|---|---|---|---|
| Jobs | | $j_2$ | $j_5, j_6, j_8$ | $j_7, j_9$ | First Child | $j_{10}, j_4, j_3, j_1$ |
| $C_i$ | 0 | 10 | 20 | 10 | | |

| Machines | $i_1$ | $i_2$ | $i_3$ | $i_4$ | | Permutation |
|---|---|---|---|---|---|---|
| Jobs | $j_5$ | $j_2$ | $j_1, j_3, j_6$ | $j_7, j_9$ | Second Child | $j_8, j_{10}, j_4$ |
| $C_i$ | 29 | 10 | 17 | 10 | | |

**d) Offspring**

| Machines | $i_1$ | $i_2$ | $i_3$ | $i_4$ | |
|---|---|---|---|---|---|
| Jobs | $j_1, j_{10}$ | $j_2, j_4$ | $j_5, j_6, j_8$ | $j_3, j_7, j_9$ | First Child |
| $C_i$ | 24 | 20 | 20 | 20 | |

| Machines | $i_1$ | $i_2$ | $i_3$ | $i_4$ | |
|---|---|---|---|---|---|
| Jobs | $j_5$ | $j_2, j_4, j_8$ | $j_1, j_3, j_6, j_{10}$ | $j_7, j_9$ | Second Child |
| $C_i$ | 29 | 30 | 17 | 10 | |

**Figure 2.** AGLX operator.

### 2.3. Download Mutation Operator

The GGA includes the Download mutation operator that uses two phases to modify two genes in a solution. Algorithm 2 contains the procedure of the Download mutation operator. In the first stage, called download, the operator clusters the genes (machines) among two sets, $W$ and $O$ (line 1). We denote $W$, $O$ = Cluster($S$) the sets derived by grouping the machines in the solution $S$, in such a way that $W$ includes the machines with a processing time $C_i$ equal to the makespan $C_{max}$, while $O$ holds the ones with an assigned processing time $C_i$ less than the makespan $C_{max}$. Next, from each set ($W$ and $O$), one machine ($w$ and $o$) is randomly selected (lines 2 and 3). We denote $i$=Pick($M$) the machine $i$ randomly selected from the set of machines $M$ with a uniform distribution. Subsequently, the jobs in the selected machines are released (line 4). We denote $S'$, $RJ[]$= Download($S$, $w$, $o$) the solution derived by releasing the jobs of the machines $o$ and $w$, which are placed in the set $RJ[]$. Finally, the arrangement of the jobs in $RJ[]$ is modified with the permute() function mentioned above, giving rise to the set $RJ'[]$ (line 5). Later, in the second stage, the released jobs are redistributed among the selected machines $w$ and $o$ with the heuristic Best() (lines 6–8). We denote $S'' = $ Best($S'$, $j$, $w$, $o$) the solution obtained by applying the Best() heuristic. For each job $j$, this heuristic calculates the equations $C_w = C_w + p_{wj}$ and $C_o = C_o + p_{oj}$, where $C_w$ and $C_o$ represent the assigned processing time of machines $w$ and $o$, respectively, and $p_{wj}$ and $p_{oj}$ the processing time required for machines $w$ and $o$ to process the job $j$. In this way, Best() assigns $j$ to the machine that generates the lowest $C_i$ value. It is important to highlight that the main difference between the reassignment heuristics Min() and Best() is that Min() re-inserts the jobs considering all the machines, while Best() re-inserts them by considering only the two selected machines $o$ and $w$.

---

**Algorithm 2** Download mutation operator

---

**Input:** A solution $S$.
**Output:** A mutated solution $S''$.

1: $W, O$= Cluster($S$);
2: $w$= Pick($W$);
3: $o$= Pick($O$);
4: $S'$, $RJ[]$= Download($S, w. o$);
5: $RJ'[]$= Permute($RJ[]$);
6: **for all** job $j \in RJ'[]$ **do**
7:     $S'' = $ Best($S', j, w, o$);
8: **end for**
9: **end process.**

---

Figure 3 describes the mutation process of the Download operator with an example that contains an initial solution for the instance presented in Figure 1a with four genes (groups). The ten jobs, from $j_1$ to $j_{10}$, are distributed among four groups, from $i_1$ to $i_4$, and the time that each group $i$ requires to process its assigned jobs from $C_1$ to $C_4$ is stored in the vector $C_i$. Figure 3a shows the result of clustering the machines with processing time $C_i$ equal to the makespan $C_{max}$ in the set $W = \{i_1\}$ and the remaining machines in set $O = \{i_2, i_3, i_4\}$. Figure 3b indicates the machines $w = i_1$ and $o = i_4$, outlined in bold, randomly selected from the sets $W$ and $O$, respectively. Figure 3c contains the solution with the selected machines to be altered, outlined in bold, downloaded by releasing their jobs and placing them in the box of released jobs $RJ$. Finally, Figure 3d shows a permutation of the jobs in $RJ$ and the result of reinserting them with the allocation heuristic Best(). The calculation of the processing time $C_i$ of each machine $i$, as well as the operations performed by the allocation heuristic Best() to assign the released jobs, can be calculated using the example instance $I$ presented in Figure 1a. As this example shows, the quality of the mutated solution is better than that of the initial solution, demonstrating the effectiveness of the Download mutation operator.

Given the following potential solution for the test instance of Figure 1a:

| Solution | | | | |
|---|---|---|---|---|
| Machines | $i_1$ | $i_2$ | $i_3$ | $i_4$ |
| Jobs | $j_1, j_2, j_{10}$ | $j_3, j_4$ | $j_5, j_6, j_8$ | $j_7, j_9$ |
| $C_i$ | 34 | 25 | 20 | 10 |

The Download mutation operator proposed by Ramos-Figueroa *et al.* [1] works as follows:

| | | | | | |
|---|---|---|---|---|---|
| **a) Machines in the sets $W$ and $O$** | $i_1$ <br> $W$ | | $i_2, i_3, i_4$ <br> $O$ | | |
| **b) Selecting machines $w$ and $o$** | Machines <br> Jobs <br> $C_i$ <br> $w$ | $i_1$ <br> $j_1, j_2, j_{10}$ <br> 34 | $i_2$ <br> $j_3, j_4$ <br> 25 | $i_3$ <br> $j_5, j_6, j_8$ <br> 20 | $i_4$ <br> $j_7, j_9$ <br> 10 <br> $o$ |
| **c) Download** | Machines <br> Jobs <br> $C_i$ | $i_1$ <br> <br> 0 | $i_2$ <br> $j_3, j_4$ <br> 25 | $i_3$ <br> $j_5, j_6, j_8$ <br> 20 | $i_4$ <br> <br> 0 | *RJ* <br> $j_1, j_2, j_7, j_9, j_{10}$ |
| **d) Reinsertion** | Machines <br> Jobs <br> $C_i$ | $i_1$ <br> $j_1, j_{10}$ <br> 24 | $i_2$ <br> $j_3, j_4$ <br> 25 | $i_3$ <br> $j_5, j_6, j_8$ <br> 20 | $i_4$ <br> $j_2, j_7, j_9$ <br> 28 | Permutation <br> $j_1, j_{10}, j_2, j_9, j_7$ |

**Figure 3.** Download mutation operator.

## 2.4. Selection and Replacement Strategies

The GGA employs an adaptation of the controlled reproduction technique proposed by Quiroz-Castellanos et al. [28], which uses an elitist approach together with two inverted rankings to give all the solutions a chance to contribute to the next generation but forcing the survival of the best solutions. The replacement strategy preserves the population diversity and the best solutions by replacing duplicated fitness individuals and the worst fitness solutions with new offspring. Algorithm 3 contains the procedure of the ranking strategy. First, this algorithm ranks the population (line 1). We denote $P' = \text{Rank}(P)$ the individuals arranged by sorting them from best to worst according to their fitness. Next, if there are solutions with repeated fitness, only one solution is kept in the ranked, and the others are placed at the end of the ordered list (line 2). We denote $P'' = \text{Rearrange}(P)$ the population rank resulting from placing the similar solutions at the end of the ordered list. Subsequently, the solutions in $P''$ are distributed among the sets $G$, $R$, and $B$ (line 3). We denote $G, R, B = \text{Distribution}(P'')$ the sets obtained by placing the ranked solutions in the sets $G$, $R$, $B$. In this way, $G$ includes the best $n_c$ solutions, where $n_c$ is a parameter to be configured that determines the number of individuals selected for the crossover process of each generation. On the other hand, the set $R$ contains the solutions in the population $P''$ without the best $n_c/2$ solutions. Finally, the set $B$ holds the best $|B|$ individuals, called elite solutions, that receive special treatment since they have the best characteristics of the population. Therefore, $|B|$ is another parameter to be configured.

---

**Algorithm 3** Ranking strategy

> **Input:** The population $P$.
> **Output:** The population rearranged en sets the $G$, $R$, and $B$.

1: $P' = \text{Rank}(P)$;
2: $P'' = \text{Rearrange}(P)$;
3: $G, R, B = \text{Distribution}(P'')$;
4: **end process.**

---

Given this solution hierarchical structure, $n_c/2$ parent solutions are randomly taken from the set $G$, and the remaining $n_c/2$ parents are randomly picked up from the solutions in the set $R$. In this way, each pair of parents is created with a parent selected from the set $G$ and the other one from the set $R$. Hence, it is necessary to validate that parent pairs do not have the same solution since some solutions can be selected more than once. After applying the crossover operator to each pair of parents, the new individuals are incorporated into the ranked population $P''$ in the following way. Half of the generated children replace the parents selected from the set $R$, and the remaining offspring replace first the solutions with repeated fitness and then those with worse fitness, i.e., the solutions at the end of the ranked population $P''$.

Once the replacement strategy is applied, the population is rearranged again with the same ranking strategy, described in Algorithm 3, to later select the best $n_m$ solutions for mutation, where $n_m$ is a parameter to be configured that determines the number of mutated solutions each generation. When applying the mutation operator, if a solution belongs to the elite group $B$, the solution is first cloned and later mutated. The clones can be entered into the population, replacing first the solutions with repeated fitness and then those with worse fitness.

## 2.5. Impact Analysis of Crossover and Mutation Rate on GGA

In order to identify the impact of each variation operator (crossover and mutation) on the GGA performance, an experimental study was performed by using three different values for the number of individuals selected for the crossover process ($n_c$) and the number of solutions to be mutated ($n_m$): 20, 40, and 60. In this way, GGA was run with the 9 configurations ($Conf$) generated from all possible combinations of these three parameters: $Conf_1$:

$n_c = 20$, $n_m = 20$, $Conf_2$: $n_c = 20$, $n_m = 40$, ... $Conf_9$: $n_c = 60$, $n_m = 60$. Figure 4 shows a bar graph of the results obtained from this study, where each bar represents 1 of the 9 configurations grouped according to the number of mutated solutions ($n_m$), and each pattern indicates the number of selected individuals for the crossover process ($n_c$): squares = 20, waves = 40, and circles = 60. As Figure 4 indicates, the GGA performance tends to improve (lower error rate) as the number of individuals considered for the crossover and mutation processes increases, although the crossover operator shows a higher impact on its performance. This behavior is different from the one presented by the GGA-CGT, where the mutation operator has the greatest positive impact on the final performance of this algorithm. The results and conclusions obtained from this study motivated the review of the mutation operator, exploring different strategies to include those that contribute to the impact improvement of this operator on the GGA final performance on solving the $R||C_{max}$ problem.

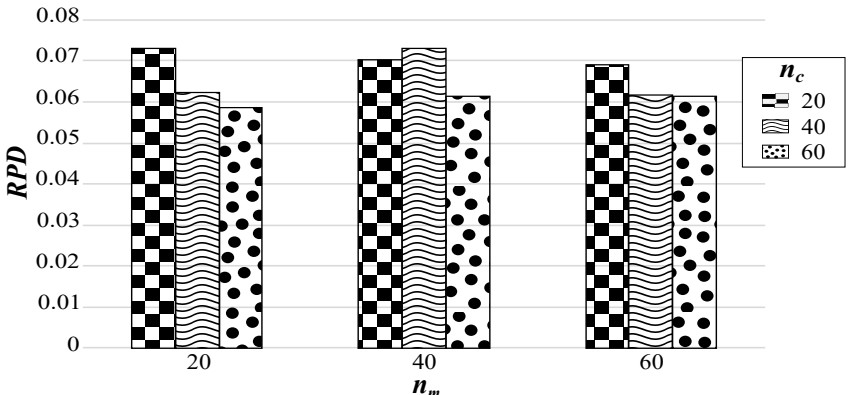

**Figure 4.** Impact analysis of the parameters: number of individuals selected for crossover $n_c$ and number of mutated solutions $n_m$ in the GGA final performance.

### 3. Grouping Mutation Operators

The mutation is a genetic operator generally used to control population diversity during the GGA search process. The mutation operators for the GGA are called grouping mutation operators since they work at the group level. That is, they select $g$ groups using some criterion (such as selecting the best, the worst, or random groups) to slightly modify them employing different operations. According to Ramos-Figueroa et al. [32], the specialized literature holds seven mutation operators designed for GGAs in addition to the Download operator. Three of them, the Swap, the Insertion, and the Item Elimination, perform small alterations in the solutions with operations directly applied to some items of the selected groups. In contrast, the remaining operators, called Elimination, Creation, Merge and Split, and Reordering, promote more severe disturbances in solutions since they perform operations involving all the items of the selected groups.

The seven mutation operators have been used to solve a wide variety of grouping problems with different conditions and constraints. Due to these differences, mutation operators must be adapted to the characteristics of the problem to be solved. As a result, grouping mutation operators for the $R||C_{max}$ problem can differ in the tactics that they use to select the jobs and machines involved in the mutation operations, the strategies employed to handle the jobs and the selected machines, and the problem-domain heuristics included. The following sections show the general procedure of four state-of-the-art grouping mutation operators: Swap, Insertion, Elimination, and Merge and Split. This study contemplates the best state-of-the-art mutation operators that apply for the $R||C_{max}$ problem, discarding the infeasible ones and those which have not shown outstanding performance. However, in [32] a more detailed description of the seven group-oriented mutation operators procedure can be found, as well as a compilation of other mutation

operators applied to different grouping problems and the parameter settings approach that they use. It is important to note that in addition to the Download operator, none of the four mutations described below have been used to solve the $R||C_{max}$ grouping problem. The above motivates this experimental study, whose main objective is to explore the performance of the most used mutation operators to solve $R||C_{max}$.

### 3.1. The Swap Operator

The Swap operator selects two groups to later pick $k$ items from each selected group and exchange the items from one group to another. Due to its way of working, it can be adapted and used to solve grouping problems with different constraints and conditions. Thanks to this quality, the Swap operator has been used to solve classic problems such as Bin Packing [34] as well as new problems such as Maximally Diverse [35].

### 3.2. The Insertion Operator

Similar to the Swap operator, the Insertion operator selects two groups to later pick $k$ items from one selected group and insert them to the other group. This operator has been used to solve from classic problems such as Graph Coloring [36] to newer problems such as Group Stock Portfolio [37], covering problems with different constraints and conditions [38].

### 3.3. The Elimination Operator

The Elimination operator chooses $g$ groups to remove them, release their items, and re-insert them by applying problem-domain heuristics, for example, the heuristic Min() used by the state-of-the-art GGA for $R||C_{max}$. According to the scope of the literature review, this is the most used mutation operator to solve grouping problems because it has shown promising results, mainly in classic problems such as Bin Packing [28], Cell Formation [39], Multiple Knapsack [40], and Timetabling [41].

### 3.4. The Merge and Split Operator

The Merge and Split, also known as Division and Combination operator, works in two phases. In the first stage, it selects two groups and transforms them into a single one. Then, in the second stage, it picks a group to distribute its items between two distinct groups. Merge and Split has been used to solve grouping problems such as Cell Formation [42] and Multivariate Micro-aggregation [43].

## 4. Computational Experiments

This section presents the experimental design proposed to analyze the way different elements involved in the mutation process can impact the performance of grouping mutation operators. The objective of this work is to design an efficient grouping mutation operator that includes the best features identified during the experimentation, to later incorporate it to the state-of-the-art GGA for $R||C_{max}$ to improve its performance [1]. The experimental design consists of four phases. The first stage covers the analysis of the state-of-the-art grouping mutation operators to determine which one has the best performance for $R||C_{max}$. The second phase comprises an exploratory analysis to observe the influence of the numbers of machines and jobs involved in mutation operations. The third phase includes the assessment of different machine selection strategies, including biased, random, and mixed approaches. Finally, the fourth phase studies the contribution of distinct rearrangement heuristics based on insertion and swap operations. The main objective of these strategies is to reorganize some jobs of the solutions applying more complex and expensive processes. Although they involve a computational cost, they are of vital importance when the mutation operator alone is unable to leave a local optimum. The information collected is used to design an efficient grouping mutation operator for $R||C_{max}$.

The performance assessment of each operator involves solving 1400 test instances introduced by Fanjul-Peyro in 2010, distributed among 7 sets [26]. The first 5 sets differ in the range employed to generate the $p_{ij}$ values with a uniform distribution: $U(1, 100)$,

$U(10, 100)$, $U(100, 120)$, $U(100, 200)$, and $U(1000, 1100)$. From the remaining sets, one includes instances with correlated machines (*MacCorr*) and the other instances with correlated jobs (*JobsCorr*). These instances can consider 100, 200, 500, or 1000 jobs and 10, 20, 30, 40, or 50 machines. Each set contains 200 instances, 10 for each combination of the number of machines $m$ and jobs $n$.

To analyze the performance of each operator, we generate a population of 100 individuals with the heuristic Min() to later mutate them for 500 generations. For a fair comparison, we use the same seed for each operator. Finally, we use the average Relative Percentage Deviation (*RPD*) to compare the operators performance. Given an instance $i$, the *RPD* is defined as in (1), where $C_{max}(i)$ depicts the $C_{max}$ value found by the operator, and $C_{max}^*(i)$ represents the best $C_{max}$ found using two hours of the commercial solver CPLEX. Thus, *RPD* indicates the deviation from the evaluated grouping mutation operators to CPLEX.

$$RPD = \frac{C_{max}(i) - C_{max}^*(i)}{C_{max}^*(i)} \tag{1}$$

### 4.1. State-of-the-Art Mutation Operators

This experiment aims to study the performance of the state-of-the-art grouping mutation operators in the problem $R||C_{max}$. Recalling from Section 3, this study comprises four operators: Swap, Insertion, Elimination, and Merge and Split, since this work focuses on the best state-of-the-art mutation operators that apply for the $R||C_{max}$ problem. However, the specialized literature contains other mutation operators applied to various grouping problems with different constraints and conditions [32]. Next, the procedures of the four mutation operators adapted to work with the constraints and conditions of the problem $R||C_{max}$ are presented. This information is reinforced by Algorithms 4–7 and Figure 5 that includes an example for each operator.

Algorithm 4 contains the procedure of the Swap mutation operator. First, it selects two machines $i_A$ and $i_B$ (lines 1 and 2). We denote $i =$ PickMachine($S$) the machine $i$ randomly selected from the solution $S$ with a uniform distribution. Later, this operator selects one job for each chosen machine (lines 3 and 4). We denote $j =$ PickJob($i$) the job $j$ randomly selected from the machine $i$ with a uniform distribution. Finally, the operator interchanges the two picked up jobs (line 5). We denote $S' =$ Interchange($S, i_A, i_B, j_A, j_B$) the solution derived by interchanging the jobs $j_A$ and $j_B$ in machines $i_A$ and $i_B$. Figure 5a explains the mutation process of the Swap operator adapted to solve the problem $R||C_{max}$ with an example in which the jobs $j_A = j_1$ and $j_B = j_7$, selected from machines $i_A = i_1$ and $i_B = i_4$, respectively, are exchanged. In this way, in the initial individual (*Solution*), the machines $i_A$ and $i_B$ outlined in bold and the jobs in bold $j_A$ and $j_B$ depict the machines and the jobs selected, respectively; and the final individual (*Mutation*) shows the jobs in their new position.

---

**Algorithm 4** Swap operator

    **Input:** A solution $S$.
    **Output:** A mutated solution $S'$.
1: $i_A =$ PickMachine($S$);
2: $i_B =$ PickMachine($S$);
3: $j_A =$ PickJob($i_A$);
4: $j_B =$ PickJob($i_B$);
5: $S' =$ Interchange($S, i_A, i_B, j_A, j_B$);
6: **end process.**

---

Given a test instance of $R||C_{max}$ with $n=9$ jobs from $j_1$ to $j_9$ and $m=4$ machines from $i_1$ to $i_4$ where each cell indicates the processing time $p_{ij}$ that each machine $i$ requires to process every job $j$, as well as a solution where each gene represents a machine $i$ that contains its assigned jobs and the processing time $C_i$ that it needs to process such jobs:

Test Instance

| M \ N | $i_1$ | $i_2$ | $i_3$ | $i_4$ |
|---|---|---|---|---|
| $j_1$ | 20 | 15 | 10 | 12 |
| $j_2$ | 10 | 12 | 16 | 18 |
| $j_3$ | 18 | 15 | 11 | 10 |
| $j_4$ | 15 | 17 | 16 | 11 |
| $j_5$ | 10 | 20 | 11 | 19 |
| $j_6$ | 20 | 12 | 15 | 16 |
| $j_7$ | 11 | 19 | 20 | 13 |
| $j_8$ | 12 | 10 | 18 | 20 |
| $j_9$ | 15 | 20 | 12 | 11 |

Solution

| Machines | $i_1$ | $i_2$ | $i_3$ | $i_4$ |
|---|---|---|---|---|
| Jobs | $j_1, j_9$ | $j_2, j_4$ | $j_5, j_8$ | $j_3, j_6, j_7$ |
| $C_i$ | 35 | 29 | 29 | 39 |

The four group-oriented mutation operators work as follows:

**a) Swap**

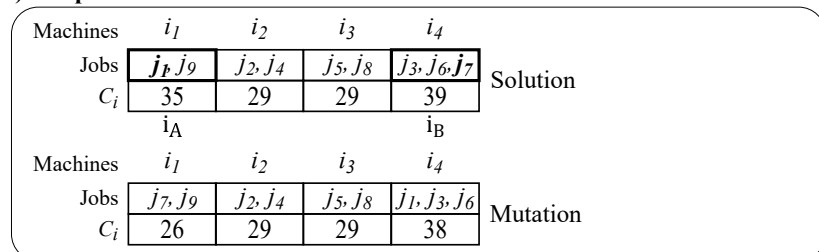

**b) Insertion**

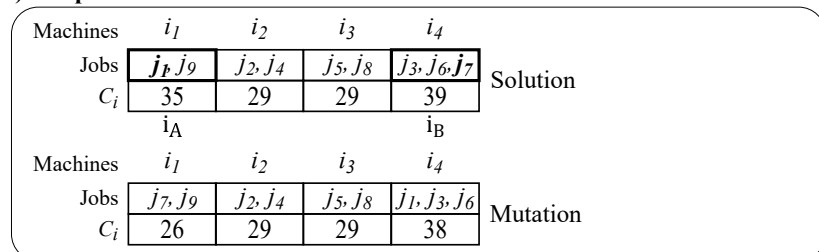

**c) Elimination**

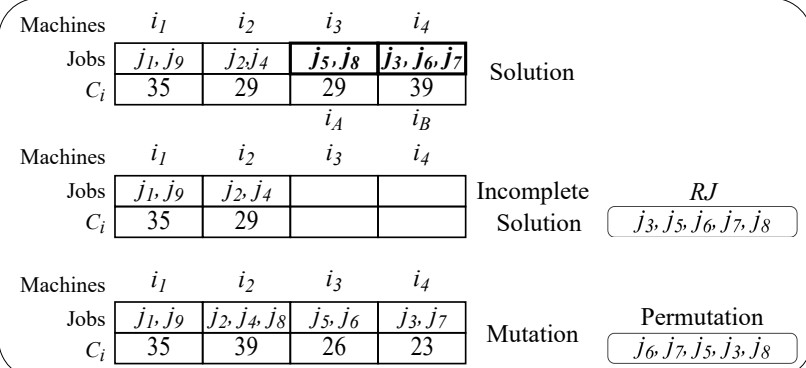

**d) Merge & Split**

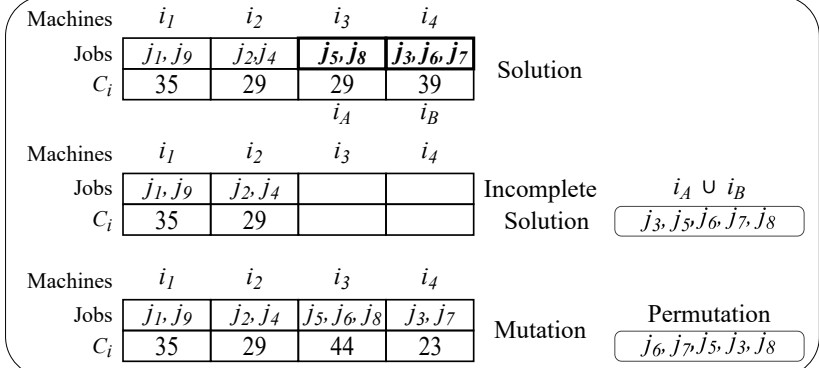

**Figure 5.** Group-oriented mutation operators adapted for $R||C_{max}$.

Similarly, Algorithm 5 includes the procedure of the Insertion mutation operator. First, it uses the before-mentioned PickMachine() function to select two machines $i_A$ and $i_B$ (lines 1 and 2). Next, it employs the PickJob() function described above to select a job $j_A$ from the first selected machine $i_A$ (line 3). Finally, this operator inserts the job $j_A$ into the second selected machine $i_B$ (line 4). We denote $S' = \text{Insertion}(S, i_A, i_B, j_A)$ the solution derived by inserting the job $j_A$ from machine $i_A$ into machine $i_B$. Figure 5b describes the mutation process of the Insertion operator implemented to solve the problem $R||C_{max}$ with an example, where the job $j_A = j_7$, selected from machine $i_A = i_4$, is inserted into machine $i_B = i_1$. For a clear explanation, the example outlines in bold the selected machines $i_A$ and $i_B$ and highlights the inserted item in bold $j_A$ in the initial individual (*Solution*). Thus, the final individual (*Mutation*) shows the picked job $j_A$ in its new position.

---

**Algorithm 5** Insertion operator

    **Input:** A solution $S$.
    **Output:** A mutated solution $S'$.
1:  $i_A$ = PickMachine($S$);
2:  $i_B$ = PickMachine($S$);
3:  $j_A$ = PickJob( $i_A$ );
4:  $S'$ = Insertion($S$, $i_A$, $i_B$, $j_A$);
5:  **end process.**

---

On the other hand, Algorithm 6 describes the procedure of the Elimination operator. Like the Swap and the Insertion operators, the Elimination process starts by picking up two machines $i_A$ and $i_B$ by using the PickMachine() function (lines 1 and 2). Next, it places all the jobs of both machines in the set of released jobs $RJ$, employing the before-mentioned Download() function (line 3). It is important to remark that this process is performed instead of the elimination, since the machines cannot be removed due to the characteristics of the problem. Subsequently, the location of the jobs in $RJ$ is modified by using the Permute() function (line 4). Finally, the permuted jobs in $RJ'[]$ are re-inserted with the Min() heuristic (lines 5–7). Figure 5c explains the mutation process of the Elimination operator adapted to solve the problem $R||C_{max}$ with an example, where the machines outlined in bold $i_A = i_3$ and $i_B = i_4$ depict the machines selected to remove their jobs $j_3$, $j_5$, $j_6$, $j_7$, and $j_8$ highlighted in bold from the initial individual (*Solution*). The *Incomplete Solution* shows the chromosome without the released items placed in the box $RJ$. Lastly, the box *Permutation* represents the jobs in $RJ$ reordered randomly, and the final solution *Mutation* depicts the chromosome generated by assigning the jobs in the box *Permutation* by using the problem-domain heuristic Min().

---

**Algorithm 6** Elimination operator

    **Input:** A solution $S$.
    **Output:** A mutated solution $S''$.
1:  $i_A$ = PickMachine($S$);
2:  $i_B$ = PickMachine($S$);
3:  $S'$, $RJ[]$ = Download($S$, $i_A$, $i_B$);
4:  $RJ'[]$ = Permute($RJ[]$);
5:  **for all** job $j \in RJ'[]$ **do**
6:     $S''$ = Min($S'$, $RJ'[]$);
7:  **end for**
8:  **end process.**

---

Lastly, Algorithm 7 contains the procedure of the Merge and Split operator. Similar to the before-described mutation operators, Merge and Split begins by choosing two machines $i_A$ and $i_B$ in a random way with the PickMachine() function (lines 1 and 2). Later, it

locates the jobs of the selected machines in the set of released jobs $RJ$ with the Download() function (line 3). As can be seen, it is a similar case to the elimination since the machines cannot be joined or split. Hence, the operator uses the Permute() function to modify the location of the jobs in $RJ$ (line 4), and later, it simulates the splitting part by re-inserting the permuted jobs in $RJ'[]$ among the two selected machines $i_A$ and $i_B$ using the above-described heuristic Best() (lines 5–7). Figure 5d includes the mutation process of the Merge and Split operator with an example that contains an initial individual (*Solution*) with the two selected machines $i_A$ and $i_B$ outlined in bold and the released jobs $j_3$, $j_5$, $j_6$, $j_7$, and $j_8$ highlighted in bold. Moreover, the example contains the *Incomplete Solution* without the jobs in $i_A \cup i_B$ placed in a box with the same name ($i_A \cup i_B$). Lastly, this figure includes the final solution *Mutation* that depicts the chromosome resulting from the allocation of the jobs in *Permutation* (a box with the jobs in $i_A \cup i_B$ reordered randomly) by applying the problem-domain heuristic Best().

---

**Algorithm 7** Merge and Split operator

---

　　**Input:** A solution $S$.
　　**Output:** A mutated solution $S''$.
　1: $i_A$ = PickMachine($S$);
　2: $i_B$ = PickMachine($S$);
　3: $S'$, $RJ[]$ = Download($S$, $i_A$, $i_B$);
　4: $RJ'[]$ = Permute($RJ[]$);
　5: **for all** job $j \in RJ'[]$ **do**
　6:　　$S'' = $ Best($S'$, $j$, $w$, $o$);
　7: **end for**
　8: **end process.**

---

Table 1 shows the results obtained from the computational experiments. For a comprehensive study, the performance of the operators was analyzed considering the number of jobs $n$, the number of machines $m$, the distribution of the processing times $p_{ij}$, and the 1400 instances together. In this way, the first column indicates the criterion used to study the performance of the operators, the second one contains the classes covered for each grouping criterion, and the following columns represent the average $RPD$ (Relative Percentage Deviation) achieved by each operator: Swap, Insertion, Merge and Split, and Elimination, respectively. Finally, this table highlights in bold the results obtained by the best operator for each group of instances. From Table 1, it can be observed that the Elimination operator excelled in all the criteria used to distribute the instances. It is important to note that the four operators had a similar performance since their average $RPD$ differs only by hundredths.

Moreover, it is remarkable that the Download mutation operator procedure of the studied GGA is quite similar to the state-of-the-art mutation operator Merge and Split, since although the operations merge and split cannot be applied to groups explicitly due to the characteristics and conditions of the problem, they can be emulated by considering the jobs. In this way, the first stage of the Download mutation operator represents the combination of the groups, where the jobs of the two selected machines are released and placed in a single set. Similarly, the second stage depicts the split operation, where the jobs are redistributed among the selected machines. Finally, it is also important to mention that the only difference between the Merge and Split operator and the Elimination operator (the two operators with the best performance) is the job reassignment strategy they work with, since Merge and Split re-inserts the jobs only on the two selected machines, while the Elimination operator tries to re-insert the jobs on all the machines.

The following stages of this experimental study contain the analysis of different aspects involved in the mutation operator with the reassignment heuristic that considers all the machines, such as the number of machines to handle, the number of jobs to remove, the machine selection strategy, and the rearrangement heuristics.

**Table 1.** Comparison of Swap, Insertion, Merge and Split, and Elimination mutation operators using RPD.

|  | Instance Set | Swap | Insertion | Merge and Split | Elimination |
|---|---|---|---|---|---|
|  | 100 | 0.1213 | 0.1219 | 0.1071 | **0.0804** |
| $n$ | 200 | 0.1408 | 0.1432 | 0.1353 | **0.1154** |
|  | 500 | 0.1365 | 0.1371 | 0.1372 | **0.1281** |
|  | 1000 | 0.1380 | 0.1381 | 0.1387 | **0.1350** |
|  | 10 | 0.1291 | 0.1290 | 0.1291 | **0.1178** |
|  | 20 | 0.1391 | 0.1402 | 0.1344 | **0.1229** |
| $m$ | 30 | 0.1256 | 0.1252 | 0.1220 | **0.1074** |
|  | 40 | 0.1310 | 0.1331 | 0.1270 | **0.1084** |
|  | 50 | 0.1460 | 0.1478 | 0.1353 | **0.1172** |
|  | $U(1, 100)$ | 0.2802 | 0.2740 | 0.2632 | **0.2107** |
|  | $U(10, 100)$ | 0.2080 | 0.2060 | 0.2039 | **0.1802** |
|  | $U(100, 120)$ | 0.0417 | 0.0438 | 0.0408 | **0.0384** |
| $P_{ij}$ | $U(100, 200)$ | 0.1230 | 0.1248 | 0.1198 | **0.1164** |
|  | $U(1000, 1100)$ | 0.0218 | 0.0230 | 0.0214 | **0.0201** |
|  | *JobsCorr* | 0.1259 | 0.1307 | 0.1194 | **0.1049** |
|  | *MacsCorr* | 0.1385 | 0.1432 | 0.1384 | **0.1326** |
|  | 1400 instances | 0.1341 | 0.1351 | 0.1296 | **0.1147** |

*4.2. Handled Machines and Removed Jobs*

After observing that the four operators of the state of the art showed quite similar performance and that the Elimination operator slightly excelled, the second phase of the experimental study focused on analyzing how the number of handled machines and removed jobs impact the performance of the mutation operator. To analyze this phenomenon, we explore thirty-five variants of the operator. This study consists of evaluating the suitability of removing 1, 2, 3, 4, 6, 8, and 10 jobs from 2, 4, 6, 8, and 10 different machines, where each combination of removed jobs and managed machines results in an operator. For this study, we designed an enhanced version of the Elimination operator, called Elimination operator-v2. Algorithm 8 contains the procedure of this version that is able to adapt itself to the number of machines $f$ and jobs $h$ to handle. Therefore, this version receives the solution and the number of machines and jobs to consider. Thus, it starts by using a cycle to select the machines with the PickMachine() function (lines 1 and 2). Furthermore, for each machine, it employs another cycle to choose the $h$ jobs with the PickJob() function and place them in the set of released jobs $RJ$ (lines 3–5). It is important to highlight that if a machine does not have enough jobs $h$, all of them are released and placed in $RJ$. Finally, the functions Permute() and Min() are used to modify the location of the jobs and re-insert them, respectively (lines 7–10).

For a fair comparison, all the operators use randomness to select the machines and the jobs that intervene in their mutation process. Thus, each operator releases $k$ jobs from $g$ machines and then re-inserts them with the heuristic Min(). As in the first phase, for each operator, 100 individuals were generated and mutated during 500 generations using the same seed.

Table 2 shows the experimental results of the thirty-five variants of the mutation operator. The first column indicates the number of machines that each operator manages, the second one represents the number of jobs removed from each of the handled machines, and the last column contains the average *RPD* of each operator for the 1400 test instances, highlighting in bold the result obtained by the best variant of the thirty-five mutation operators.

---

**Algorithm 8** Elimination operator-v2

---

**Input:** A solution $S$, number of machines $f$ and jobs $h$ handle.

**Output:** A mutated solution $S''$.

1: **for each** machine $i$ from 1 to $f$ **do**
2:      $i$= PickMachine($S$);
3:      **for each** job $j$ from 1 to $h$ **do**
4:          $RJ[]$ = PickJob($i$);
5:      **end for**
6: **end for**
7: $RJ'[]$= Permute($RJ[]$);
8: **for all** job $j \in RJ'[]$ **do**
9:      $S'' = \text{Min}(S', RJ'[])$;
10: **end for**
11: **end process.**

---

**Table 2.** Comparison of handled machines and removed jobs using *RPD*.

| Handled Machines | Removed Jobs | RPD |
|:---:|:---:|:---:|
| | 1 | **0.091437** |
| | 2 | 0.094475 |
| | 3 | 0.097644 |
| 2 | 4 | 0.100010 |
| | 6 | 0.102259 |
| | 8 | 0.103456 |
| | 10 | 0.103984 |
| | 1 | 0.093067 |
| | 2 | 0.100647 |
| | 3 | 0.104505 |
| 4 | 4 | 0.107246 |
| | 6 | 0.109475 |
| | 8 | 0.111302 |
| | 10 | 0.111263 |
| | 1 | 0.095776 |
| | 2 | 0.105834 |
| | 3 | 0.109519 |
| 6 | 4 | 0.111925 |
| | 6 | 0.114454 |
| | 8 | 0.115151 |
| | 10 | 0.115754 |
| | 1 | 0.09889 |
| | 2 | 0.109016 |
| | 3 | 0.112681 |
| 8 | 4 | 0.114861 |
| | 6 | 0.116797 |
| | 8 | 0.117525 |
| | 10 | 0.117800 |
| | 1 | 0.102228 |
| | 2 | 0.110804 |
| | 3 | 0.114677 |
| 10 | 4 | 0.116184 |
| | 6 | 0.117627 |
| | 8 | 0.118031 |
| | 10 | 0.117819 |

It appears from Table 2 that the operators that release only one job from each machine perform better than those that release more and that the best option is to consider only two machines. Moreover, to graphically observe the behavior of the 35 designed operators, the 1400 instances were grouped into 20 groups concerning each combination of jobs (100, 200, 500, and 1000) and machines (10, 20, 30, 40, and 50) to calculate the average *RPD* of each group and analyze the impact of each operator in more detail, e.g., the group where $m = 10$ and $n = 100$, the group where $m = 10$ and $n = 200$, and so on. Figures 6 and 7 contain

two representative graphs of the behavior presented by the thirty-five mutation operator variants, which allow observing the impact of the two evaluated features, i.e., the number of machines to be handled and the number of jobs to be removed from each machine.

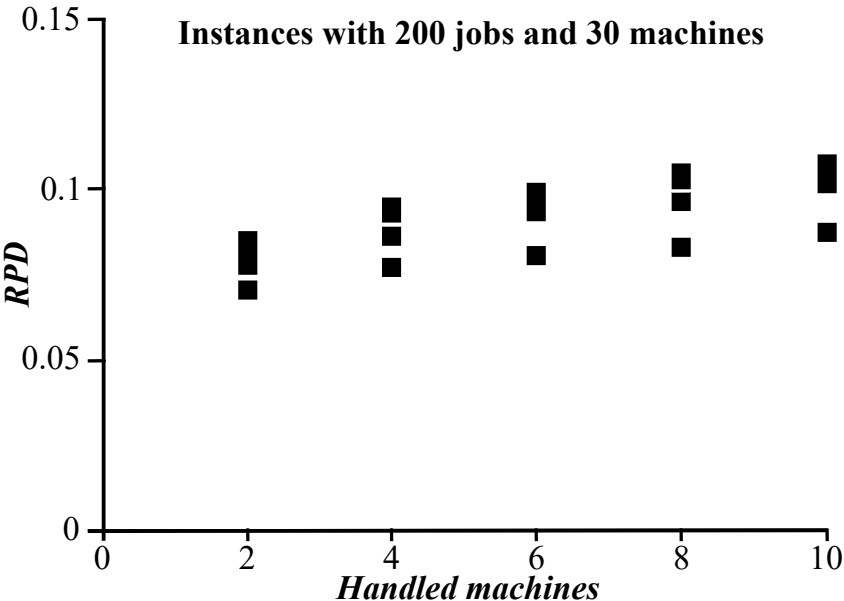

**Figure 6.** Behavior of the mutation operators grouped by the number of handled machines.

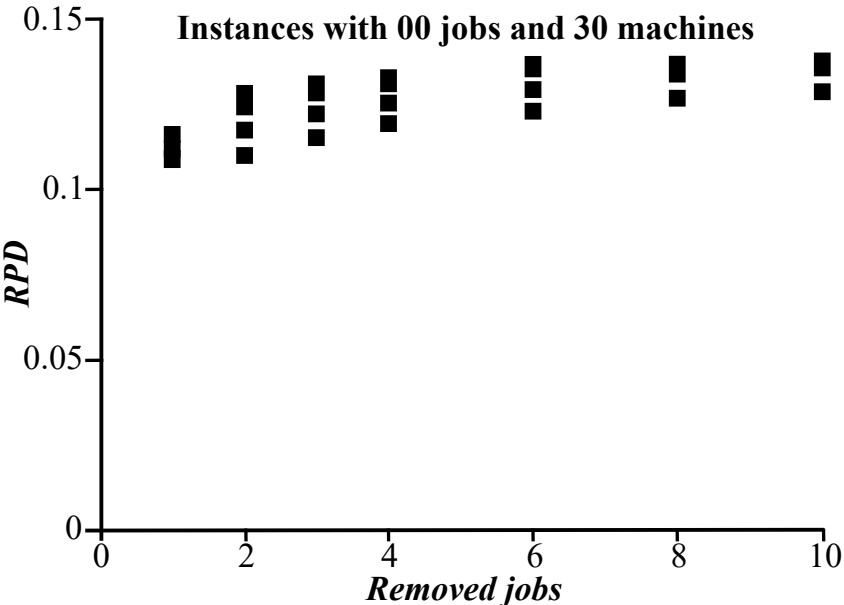

**Figure 7.** Behavior of the mutation operators grouped by the number of removed jobs from each handled machine.

Figure 6 allows observing the behavior of the operator's performance grouped according to the number of machines that they handle for all instances with 200 jobs and 30 machines. The $x$-axis of this figure indicates the number of machines handled, and the $y$-axis contains the average $RPD$ reached for each operator. On the other hand, Figure 7 groups the operators according to the number of jobs removed from each machine in instances with 500 jobs and 20 machines. The $x$-axis contains the operators grouped according to the number of jobs that they remove, and the $y$-axis contains the average $RPD$ reached for each operator. In this way, Figure 6 allows graphically observing that the performance of the operators improves as the number of handled machines decreases, while Figure 7

shows that the operators removing fewer jobs have better performance. In this fashion, the analysis suggests that the operators handling a fewer number of machines and releasing fewer jobs are more suitable.

### 4.3. Machines Selection Strategy

Once identifying that the variant that considers two machines and releasing one job from each machine has the best performance, in this stage, we evaluate the performance of four machine selection strategies, Random, Worst, Worst Best, and Worst Random, to analyze how they affect the performances of the mutation operators. Given a solution to be mutated, these strategies work as follows. Random chooses the two machines randomly. Worst selects the two machines with the worst $C_i$ values (i.e., the machines with the highest loads). Worst Best picks the worst and the best machine (i.e., the machines with the highest and the lowest loads). If there are several machines with the lowest or highest load, first, they are identified to later use a uniform distribution to select one of them randomly. Finally, Worst Random divides the machines into two groups ($W$ and $O$) in such a way that $W$ contains the machines with $C_i = C_{max}$ and $O$ the remaining machines. Next, it randomly selects the machines $w$ and $o$ from sets $W$ and $O$, respectively. It is important to note that for each machine selection strategy, the two released jobs are selected randomly using a uniform distribution and later re-inserted employing the heuristic Min().

Table 3 shows the experimental results of the operators with the four machine selection strategies. As can be seen, this table has the same structure as Table 1. That is, it clusters the instances according to the number of jobs $n$, the number of machines $m$, the distribution of the processing times $p_{ij}$ of the instances, and the 1400 test instances together. Therefore, the first column indicates the criterion used to study the performance of the operators, the second one contains the classes covered for each grouping criterion, and the following columns represent the average $RPD$ (Relative Percentage Deviation) achieved by the operators with each machine selection strategy: Random, Worst, Worst Best, and Worst Random. Finally, this table highlights in bold the results obtained by the best mutation operator for each group of instances. The experimental results in Table 3 suggest that the most suitable machine selection strategy is Worst Random, with an average $RPD$ of 0.0674 since the other approaches (Random, Worst, and Worst Best) reached higher $RPD$ averages of 0.0913, 0.0875, and 0.0912, respectively.

**Table 3.** Comparison of mutation operators with selection strategies Random, Worst, Worst Best, and Worst Random using $RPD$.

| | Instance Set | Random | Worst | Worst Best | Worst Random |
|---|---|---|---|---|---|
| $n$ | 100 | 0.0605 | 0.0577 | 0.0618 | **0.0296** |
| | 200 | 0.0848 | 0.0797 | 0.0832 | **0.0533** |
| | 500 | 0.1030 | 0.0987 | 0.1028 | **0.0827** |
| | 1000 | 0.1175 | 0.1147 | 0.1178 | **0.1046** |
| $m$ | 10 | 0.0873 | 0.0857 | 0.0894 | **0.0718** |
| | 20 | 0.0978 | 0.0942 | 0.0977 | **0.0752** |
| | 30 | 0.0842 | 0.0824 | 0.0854 | **0.0635** |
| | 40 | 0.0908 | 0.0853 | 0.0885 | **0.0631** |
| | 50 | 0.0963 | 0.0900 | 0.0951 | **0.0634** |
| $P_{ij}$ | $U(1, 100)$ | 0.1430 | 0.1470 | 0.1522 | **0.1146** |
| | $U(10, 100)$ | 0.1321 | 0.1319 | 0.1362 | **0.1003** |
| | $U(100, 120)$ | 0.0351 | 0.0309 | 0.0329 | **0.0244** |
| | $U(100, 200)$ | 0.1017 | 0.0939 | 0.0970 | **0.0740** |
| | $U(1000, 1100)$ | 0.0182 | 0.0155 | 0.0171 | **0.0123** |
| | *JobsCorr* | 0.0909 | 0.0820 | 0.0810 | **0.0576** |
| | *MacsCorr* | 0.1179 | 0.1112 | 0.1220 | **0.0888** |
| | 1400 instances | 0.0913 | 0.0875 | 0.0912 | **0.0674** |

### 4.4. Rearrangement Heuristics

After identifying the machine selection strategy that provides the best performance to the mutation operator, we noted that there are high possibilities that the genetic material of many solutions does not undergo any alteration during the mutation process. Such a phenomenon can occur because it is likely that the two released jobs can be re-inserted in the same machine to which they belonged. In order to analyze the above, we evaluated the success rate (i.e., the number of the alterations in the genetic material divided by the mutation attempts) of the mutation operator with the best properties identified in the two previous stages. The experimental results revealed that only about the 42% of the mutation attempts are successful.

The above motivates this stage of the experimental study that consists of evaluating the utility of incorporating two rearrangement heuristics, called Insertion and Assemble, to increase the operator's success rate and improve its performance. These heuristics are only used if, after releasing and reinserting the jobs, the genetic material of the mutated solution has not been altered.

The rearrangement heuristic Insertion seeks to reduce the number of jobs in one of the two selected machines by trying to insert each of their jobs into the other ones. Algorithm 9 has the procedure of the rearrangement heuristic Insertion. We denote $S' = $ Insertion($S$, $j_{sm}$, $sm$, $i$) the solution derived from $S$ by inserting job $j_{sm}$ ($j_w$ or $j_o$) from the selected machine $sm$ ($w$ or $o$) into machine $i$. As can be seen, this heuristic goes through the jobs $j_w$ and $j_o$ of the machines $w$ and $o$ selected with the machine selection strategy Worst Random (line 1). Thus, for each pair of jobs ($j_w$ and $j_o$), this algorithm traverses the $m$ machines (line 2). In this way, for each machine $i$ different from machine $w$ and $o$ (line 3 and line 9), it tries to insert the job $j_w$ of the worst machine $w$ (line 3) and then the job $j_o$ from the other machine $o$ (line 7) following two conditions, denoted as Cnd_1 and Cnd_2.

---

**Algorithm 9** Rearrangement heuristic Insertion

**Input:** A solution $S$ and two machines $w$ and $o$.
**Output:** A mutated solution $S'$.

1: **for all** job $j_w \in w$ & $j_o \in o$ **do**
2:      **for** machine $i$ in $S$ **do**
3:          **if** $i \mathrel{!=} w$ **then**
4:              **if** Cnd_1($S$, $j_w$, $w$, $i$) and Cnd_2($S$, $j_w$, $w$, $i$) **then**
5:                  $S' = $ Insertion ($S$, $j_w$, $w$, $i$);
6:                  **end process;**
7:              **end if**
8:          **end if**
9:          **if** $i \mathrel{!=} o$ **then**
10:             **if** Cnd_1($S$, $j_o$, $o$, $i$) and Cnd_2($S$, $j_o$, $o$, $i$) **then**
11:                $S' = $ Insertion ($S$, $j_o$, $o$, $i$);
12:                **end process;**
13:             **end if**
14:          **end if**
15:      **end for**
16: **end for**

---

Cnd_1($S$, $j_{sm}$, $sm$, $i$) (line 4 and line 10) allows verifying that the mutated solution ($S'$) will have equal or better quality than the initial solution ($S$). In this way, Cnd_1 checks out that the sum of the processing time resulted from the insertion in the intervened machines $i$ and $sm$ ($w$ or $o$) will be less than or equal to the sum of their processing times without performing the insertion. Hence, for each job $j_w$, Cnd_1($S$, $j_w$, $w$, $i$) returns TRUE if $C_w - p_{wj_w} + C_i + p_{ij_w} \leq C_w + C_i$, where $C_w$ and $C_i$ represent the time that machines $w$ and $i$ require to process their assigned jobs, respectively, while $p_{wj_w}$ and $p_{ij_w}$ depict the processing time that machines $w$ and $i$ require to process job $j_w$, respectively. Otherwise, it returns

FALSE. In the same way, for each job $j_o$, Cnd_1($S$, $j_o$, $o$, $i$) returns TRUE if $C_o - p_{oj_o} + C_i + p_{ij_o} \leq C_o + C_i$, where $C_o$ and $C_i$ represent the time that machines $o$ and $i$ require to process their assigned jobs, respectively; while $p_{oj_o}$ and $p_{ij_o}$ depict the processing time that machines $o$ and $i$ require to process job $j_o$, respectively. Otherwise, it returns FALSE.

On the other hand, Cnd_2($S$, $j_{sm}$, $sm$, $i$) (line 4 and line 10) checks out that the mutated solution ($S'$) will have equal or better quality than the initial solution ($S$). Cnd_2 verifies that the processing time $C_i$ of the machine $i$ with the new job, either $j_w$ or $j_o$, will be less than or equal to the current makespan $C_{max}$. Therefore, for each job $j_w$, Cnd_2($S$, $j_w$, $w$, $i$) returns TRUE if $C_i + p_{ij_w} \leq C_{max}$. Otherwise, it returns FALSE. Similarly, for each job $j_o$, Cnd_2($S$, $j_o$, $o$, $i$) returns TRUE if $C_i + p_{ij_o} \leq C_{max}$. Otherwise, it returns FALSE.

In this way, the function Insertion($S$, $j_{sm}$, $sm$ , $i$) (lines 5 and 11) is applied to $S$ if and only if a job $j$ ($j_w$ or $j_o$) satisfies the two conditions (Cnd_1 and Cnd_2). The rearrangement process ends once an insertion is performed (lines 6 and 12), but if none of the jobs satisfied the three conditions, the mutated solution would remain with its genetic material without any modification.

On the other hand, the rearrangement heuristic Assemble uses two functions. The first one is the Insertion($S$, $j_{sm}$, $sm$, $i$) that works similarly to the above rearrangement heuristic. Additionally, it incorporates a second function called Interchange that seeks to exchange each job of the selected machines with each job of the other machines in an attempt to reduce the processing time of the selected machines. Algorithm 10 contains the procedure of the rearrangement heuristic Assemble. We denote $S' = $ Interchange($S$, $j_{sm}$, $sm$, $j_i$, $i$) the solution derived from $S$ by exchanging job $j_{sm}$ ($j_w$ and $j_o$) from the selected machine $sm$ ($w$ or $o$) with each job $j_i$ in machine $i$. Like the Insertion rearrangement heuristic, Assemble loops through the jobs $j_w$ and $j_o$ of the machines $w$ and $o$ selected with the machine selection strategy Worst Random (line 1). Thus, for each pair of jobs ($j_w$ and $j_o$), this algorithm goes through the $m$ machines (line 2). In this fashion, first, it tries to insert the jobs $j_w$ of the worst machine $w$ and $j_o$ of the other machine $o$ into every machine $i$ different from machines $w$ and $o$ (line 3 and line 9) according to the two conditions described in Algorithm 9: Cnd_1 and Cnd_2 (line 4 and line 10). Next, it attempts to interchange the same jobs $j_w$ and $j_o$ with each job $j_i$ in every machine $i$ (line 15) different from machine $w$ and $o$ (line 16 and line 22), validating two conditions: Cnd_3 and Cnd_4 (line 17 and line 23).

Cnd_3($S$, $j_{sm}$, $sm$, $j_i$, $i$) (line 17 and line 23) allows verifying that the mutated solution ($S'$) will have equal or better quality than the initial solution ($S$). In this way, Cnd_3 checks out that the processing time resulted from the exchange in the intervened machines $i$ and $sm$ ($w$ or $o$) will be less than or equal to the sum of their processing times without swapping their jobs. Hence, for each job $j_w$, Cnd_3($S$, $j_w$, $w$, $j_i$, $i$) returns TRUE if $(C_w - p_{wj_w} + p_{wj_i}) + (C_i - p_{ij_i} + p_{ij_w}) \leq C_w + C_i$, where $C_w$ and $C_i$ represent the time that machines $w$ and $i$ require to process their assigned jobs, respectively; $p_{wj_w}$ and $p_{ij_i}$ depict the processing time that machines $w$ and $i$ require to process jobs $j_w$ and $j_i$, respectively; $p_{wj_i}$ $p_{ij_w}$ indicate the processing time that machines $w$ and $i$ require to process job $j_i$ and $j_w$, respectively. Otherwise, it returns FALSE. In the same way, for each job $j_o$, Cnd_3($S$, $j_o$, $o$, $j_i$, $i$) returns TRUE if $(C_o - p_{oj_o} + p_{oj_i}) + (C_i - p_{ij_i} + p_{ij_o}) \leq C_o + C_i$, where $C_o$ and $C_i$ represent the time that machines $o$ and $i$ require to process their assigned jobs, respectively; $p_{oj_o}$ and $p_{ij_i}$ depict the processing time that machines $o$ and $i$ require to process jobs $j_o$ and $j_i$, respectively; and $p_{oj_i}$ and $p_{ij_o}$ indicate the processing time that machines $o$ and $i$ require to process job $j_i$ and $j_o$, respectively. Otherwise, it returns FALSE.

On the other hand, the condition Cnd_4($S$, $j_{sm}$, $sm$, $j_i$, $i$) (line 17 and line 23) validates that the processing time resulting from the interchange in the intervened machines $i$ and $sm$ ($w$ or $o$) will be less than or equal to the current makespan ($C_{max}$) of the initial solution $S$. Hence, for each job $j_w$, Cnd_4($S$, $j_w$, $w$, $j_i$, $i$) returns TRUE if ($C_w - p_{wj_w} + p_{wj_i} \leq C_{max}$) and ($C_i - p_{ij_i} + p_{ij_w} \leq C_{max}$). Otherwise, it returns FALSE. Similarly, for each job $j_o$, Cnd_4($S$, $j_o$, $o$, $j_i$, $i$) returns TRUE if ($C_o - p_{oj_o} + p_{oj_i} \leq C_{max}$) and ($C_i - p_{ij_i} + p_{ij_o} \leq C_{max}$). Otherwise, it returns FALSE.

---

**Algorithm 10** Rearrangement heuristic Assemble

---

    **Input:** A solution $S$ and two machines $w$ and $o$.

    **Output:** A mutated solution $S'$.

1:  **for all** job $j_w \in w$ & $j_o \in o$ **do**

2:     **for** machine $i$ in $S$ **do**

3:        **if** $i \mathrel{!=} w$ **then**

4:          **if** Cnd_1($S, j_w, w, i$) and Cnd_2($S, j_w, w, i$) **then**

5:            $S' =$ Insertion ($S, j_w, w, i$);

6:            **end process;**

7:          **end if**

8:        **end if**

9:        **if** $i \mathrel{!=} o$ **then**

10:          **if** Cnd_1($S, j_o, o, i$) and Cnd_2($S, j_o, o, i$) **then**

11:            $S' =$ Insertion ($S, j_o, o, i$);

12:            **end process;**

13:          **end if**

14:        **end if**

15:        **for** job $j_i$ in $i$ **do**

16:          **if** $i \mathrel{!=} w$ **then**

17:            **if** Cnd_3($S, j_w, w, j_i, i$) and Cnd_4($S, j_w, w, j_i, i$) **then**

18:              $S' =$ Interchange ($S, j_w, w, j_i, i$);

19:              **end process;**

20:            **end if**

21:          **end if**

22:          **if** $i \mathrel{!=} o$ **then**

23:            **if** Cnd_3($S, j_o, o, j_i, i$) and Cnd_4($S, j_o, o, j_i, i$) **then**

24:              $S' =$ Interchange ($S, j_o, o, j_i, i$);

25:              **end process;**

26:            **end if**

27:          **end if**

28:        **end for**

29:     **end for**

30: **end for**

---

    The Assemble process ends once an operation, either the insertion or the interchange, is accomplished (lines 6, 12, 19, and 25). If none of the jobs met the two conditions, the mutated solution remains with its genetic material without any modification.

    In this way, two variants of the operator with the best characteristics identified in the two previous stages (i.e., removing one job from two machines selected with the strategy Worst Random and re-inserting such jobs with the Min() heuristic) were created, one for each rearrangement heuristics presented in this section: Insertion and Assemble. The performance of the two variants, called Insertion and Assemble, was evaluated using the methodology mentioned above, i.e., starting from an initial population of 100 individuals that are subsequently mutated during 500 generations and using the same seed. Table 4 holds the experimental results obtained by the two mutation operators generated in this phase. Moreover, Table 4 includes the performance of the Download mutation operator, the original GGA operator described in Section 2.5, to compare the degree of improvement provided by the variants of the operator proposed in this section. For a comprehensive analysis, the performance of the operators was analyzed clustering the instances with the criteria used in the previous stages: number of jobs $n$, number of machines $m$, distribution of processing times $p_{ij}$, and the 1400 instances together. Thus, each column shows the performance of each assessed operator for the different criteria used to group the instances, highlighting in bold the results obtained by the best mutation operator.

    As can be observed in Table 4, the best variant is that with the rearrangement heuristic Assemble, which for each pair of jobs first tries the insertion and then the interchange. The

variants with the rearrangement heuristics Insertion and Assemble reached an average *RPD* of 0.0552 and 0.0395, respectively. However, it is important to note that the two versions of the mutation operators presented in this section outperformed the original Download mutation operator of the GGA studied that reached an average *RPD* of 0.1139, as well as the four state-of-the-art operators, which had an average *RPD* above 0.1.

**Table 4.** Comparison of mutation operators with the rearrangement heuristics Insertion and Assemble and the Download operator using *RPD*.

|  | Instance Set | Insertion | Assemble | Download |
|---|---|---|---|---|
| $n$ | 100 | 0.0306 | **0.0185** | 0.0730 |
|  | 200 | 0.0480 | **0.0280** | 0.1125 |
|  | 500 | 0.0631 | **0.0441** | 0.1328 |
|  | 1000 | 0.0793 | **0.0671** | 0.1383 |
| $m$ | 10 | 0.0612 | **0.0416** | 0.1261 |
|  | 20 | 0.0617 | **0.0429** | 0.1258 |
|  | 30 | 0.0497 | **0.0366** | 0.1076 |
|  | 40 | 0.0507 | **0.0376** | 0.1054 |
|  | 50 | 0.0528 | **0.0382** | 0.1048 |
| $P_{ij}$ | $U(1, 100)$ | 0.0523 | **0.0407** | 0.2307 |
|  | $U(10, 100)$ | 0.0538 | **0.0331** | 0.1862 |
|  | $U(100, 120)$ | 0.0286 | **0.0176** | 0.0358 |
|  | $U(100, 200)$ | 0.0750 | **0.0362** | 0.1072 |
|  | $U(1000, 1100)$ | 0.0150 | **0.0100** | 0.0182 |
|  | *JobsCorr* | 0.0664 | **0.0654** | 0.0892 |
|  | *MacsCorr* | 0.0952 | **0.0728** | 0.1304 |
|  | 1400 instances | 0.0552 | **0.0394** | 0.1139 |

## 5. Comparing GGA with the Old and the New Mutation Operators

Given the knowledge gained from the experimental study, we propose a mutation operator called 2-Items Reinsertion. This operator randomly chooses two jobs from two different machines selected with the strategy Worst Random to release them and later reinsert them with the allocation heuristic Min(). Furthermore, it employs the rearrangement heuristic Assemble, based on insertion and interchange operations. The rearrangement process is only applied if, after releasing and reinserting the jobs, the genetic material of the mutated solution has not been modified.

To assess the 2-Items Reinsertion mutation operator performance, we run two variants of the state-of-the-art GGA for $R||C_{max}$ [1]. One with the old mutation operator (the Download mutation operator), i.e., the state-of-the-art GGA and the Enhanced GGA (EGGA) that uses the 2-Items Reinsertion mutation instead of the Download operator to evaluate their performance over the 1400 benchmark instances. For an equivalent comparison, the effectiveness and efficiency of both GGA variants were compared by using the same parameter configuration, i.e., the one proposed by Ramos-Figueroa et al. [1]. Table 5 contains the parameter values utilized for the population size $|P|$, number of individuals selected for the crossover $n_c$, number of individuals selected for the mutation $n_m$, elite population size $|B|$, and maximal number of generations *max_gen*. In this way, we analyze the strengths and weaknesses of the 2-Items Reinsertion mutation operator, distinguishing the quality of the solutions found by each GGA variant, their search time, as well as their ability to escape from local optima.

**Table 5.** Parameter configuration.

| Parameter | Value |
|---|---|
| $|P|$ | 100 |
| $n_c$ | 20 |
| $n_m$ | 83 |
| $|B|$ | 20 |
| *max_gen* | 500 |

For a fair comparison, both algorithms were programmed in the Rust language and were compiled using Visual Studio in the 64-bits mode. The experiments were performed on a computer with an Intel Core i5 (3.10 GHz), and 16 GB in RAM. Similar to Ramos-Figueroa et al. [1], for each instance, a single execution of the algorithms was run, with the same initial seed for the random number generation.

### 5.1. Comparing the effectiveness of GGA with the old and the new mutation operators

To measure the effectiveness of the designed 2-Items Reinsertion mutation operator, we applied the two GGA variants to the 1400 test instances and measured the improvement degree in the quality of the solutions found by each algorithm based on the *RPD*. Table 6 contains the experimental results. The first and second columns indicate the criteria used to group the test instance based on the number of jobs $n$, the number of machines $m$, the processing time distribution $p_{ij}$, and the 1400 instances together. On the other hand, the remaining columns contain the average *RPD* obtained by each metaheuristic algorithm for the four grouping criteria, respectively. Finally, this table highlights in bold the results obtained by the best GGA for each group of instances.

**Table 6.** Comparison of the state-of-the-art GGA and the EGGA presented in this work using *RPD*.

|  | Instance Set | GGA | EGGA |
|---|---|---|---|
| $n$ | 100 | 0.0391 | **0.0176** |
|  | 200 | 0.0565 | **0.0224** |
|  | 500 | 0.0665 | **0.0291** |
|  | 1000 | 0.0724 | **0.0441** |
| $m$ | 10 | 0.0512 | **0.0220** |
|  | 20 | 0.0606 | **0.0306** |
|  | 30 | 0.0559 | **0.0275** |
|  | 40 | 0.0596 | **0.0308** |
|  | 50 | 0.0657 | **0.0306** |
| $P_{ij}$ | $U(1, 100)$ | 0.0719 | **0.0465** |
|  | $U(10, 100)$ | 0.0853 | **0.0361** |
|  | $U(100, 120)$ | 0.0278 | **0.0092** |
|  | $U(100, 200)$ | 0.0820 | **0.0229** |
|  | $U(1000, 1100)$ | 0.0131 | **0.0036** |
|  | *JobsCorr* | 0.0522 | **0.0380** |
|  | *MacsCorr* | 0.0780 | **0.0419** |
|  | 1400 instances | 0.0586 | **0.0283** |

Table 6 illustrates that the EGGA showed a better performance than the state-of-the-art GGA using any criteria to group the test instances. Furthermore, it is worth noting that the EGGA reaches an average *RPD* considerably lower than the state-of-the-art GGA by solving the 1,400 test instances, with 0.028 and 0.059, respectively. Additionally, we applied the Wilcoxon rank-sum test to assess whether the differences in the *RPD* achieved by both GGAs for the 1,400 test instances are statistically significant. The Wilcoxon rank-sum is a non-parametric test that compares two algorithms without assuming a normal distribution, even for small sample sizes [44]. Table 7 presents the results obtained by the Wilcoxon rank-sum for the *RPD* values reached by both algorithms in the benchmark considered with a 95%-confidence level. For a comprehensive comparison, we generated a hypothesis test for the *RPD* achieved by both GGAs in groups of instances sorted according to the number of jobs $n$, the number of machines $m$, the distribution of the processing times $p_{ij}$ of the instances, and the complete benchmark (1400 instances). In this way, the first column indicates the criterion used to compare the algorithms, the second one contains the classes covered for each grouping criterion, and the last column indicates the *p*-values obtained by the Wilcoxon test.

**Table 7.** *p*-values for the Wilcoxon test for GGA and EGGA.

|  | Instance | *p*-Value |
|---|---|---|
| *n* | 100 | $7.10 \times 10^{-26}$ |
|  | 200 | $6.70 \times 10^{-43}$ |
|  | 500 | $2.30 \times 10^{-46}$ |
|  | 1000 | $8.16 \times 10^{-29}$ |
| *m* | 10 | $4.57 \times 10^{-39}$ |
|  | 20 | $1.63 \times 10^{-29}$ |
|  | 30 | $2.98 \times 10^{-21}$ |
|  | 40 | $3.51 \times 10^{-20}$ |
|  | 50 | $3.37 \times 10^{-25}$ |
| $P_{ij}$ | $U(1, 100)$ | $1.13 \times 10^{-12}$ |
|  | $U(10, 100)$ | $1.59 \times 10^{-49}$ |
|  | $U(100, 120)$ | $2.03 \times 10^{-33}$ |
|  | $U(100, 200)$ | $1.01 \times 10^{-51}$ |
|  | $U(1000, 1100)$ | $2.25 \times 10^{-37}$ |
|  | *JobsCorr* | $2.19 \times 10^{-15}$ |
|  | *MacsCorr* | $4.44 \times 10^{-27}$ |
|  | 1400 instances | $5.44 \times 10^{-120}$ |

Table 7 indicates that the EGGA is indeed statistically better than the state-of-the-art GGA considering the *RPD* that they reached for the test benchmark for all the groups of instances considered since all *p*-values are less than the level of significance $\alpha = 0.05$.

Finally, in order to graphically show the suitability of the designed mutation operator, the experimental study presented in Section 2.5 was repeated but this time for the impact analysis of crossover and mutation rates on the EGGA. In this way, the EGGA that incorporates the 2-Items Reinsertion mutation operator was run with the same 9 configurations, i.e., $Conf_1$: $n_c = 20$, $n_m = 20$, $Conf_2$: $n_c = 20$, $n_m = 40$, ... $Conf_9$: $n_c = 60$, $n_m = 60$. Figure 8 presents a bar graph with the results obtained from this study, where each bar depicts one of the 9 configurations grouped according to the number of mutated solutions ($n_m$), and each pattern indicates the number of selected individuals for the crossover process ($n_c$): squares $= 20$, waves$= 40$, and circles$= 60$. As Figure 8 indicates, the EGGA performance is mainly related to the number of individuals considered for the mutation processes $n_m$ in such a way that the performance of the EGGA improves (lower *RPD*) as the number of mutated solutions increases. Similarly, as the number of selected individuals for the crossover process $n_c$ increases, the GGA performance improves but to a lesser degree.

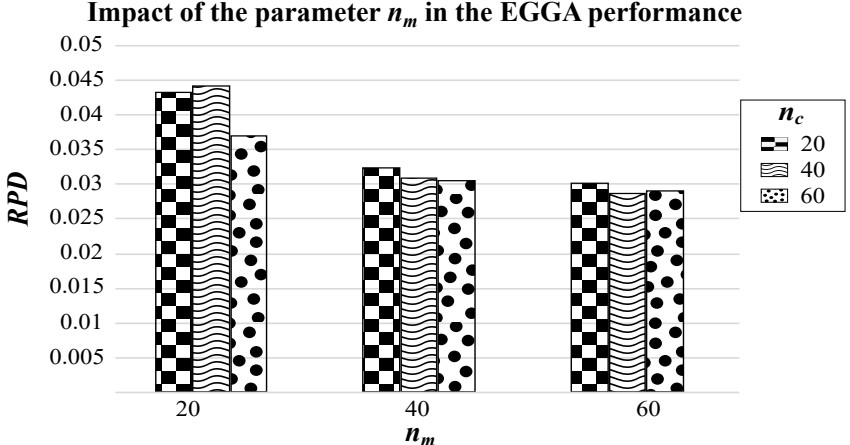

**Figure 8.** Impact analysis of the parameters: number of individuals selected for crossover $n_c$ and number of mutated solutions $n_m$ in the EGGA final performance.

The behavior mentioned above shows the suitability of the 2-Items Reinsertion mutation, which is the operator with the biggest impact on EGGA final performance and

improves it considerably. Thus, the EGGA behavior is quite similar to the one presented by the GGA-CGT [28], where the mutation operator has the greatest positive impact on the final performance of this algorithm.

*5.2. Comparing the Efficiency of GGA with the Old and the New Mutation Operators*

After analyzing the effectiveness of the EGGA, we evaluate the implications associated with the computational time of using the 2-Items Reinsertion mutation operator. Table 8 includes the experimental results. Like Table 6, the first and second columns describe the characteristics used to cluster the instances: the number of jobs $n$ and machines $m$, the processing time distribution $p_{ij}$, and the 1400 instances together. Thus, the following columns contain the average time in seconds obtained by the state-of-the-art GGA and the EGGA for each instance set, respectively.

**Table 8.** Comparison of the state-of-the-art GAA and the EGGA based on the time (time in seconds).

|  | Instance | GGA | EGGA |
|---|---|---|---|
| $n$ | 100 | 1.2 | 5.71 |
|  | 200 | 1.2 | 5.68 |
|  | 500 | 1.24 | 5.49 |
|  | 1000 | 1.36 | 9.44 |
| $m$ | 10 | 1.26 | 8.71 |
|  | 20 | 1.24 | 7.66 |
|  | 30 | 1.21 | 6.94 |
|  | 40 | 1.19 | 6.33 |
|  | 50 | 1.17 | 5.79 |
| $P_{ij}$ | $U(1, 100)$ | 1.25 | 34.09 |
|  | $U(10, 100)$ | 1.25 | 14.04 |
|  | $U(100, 120)$ | 1.25 | 2.52 |
|  | $U(100, 200)$ | 1.25 | 2.88 |
|  | $U(1000, 1100)$ | 1.25 | 2.71 |
|  | *JobsCorr* | 1.25 | 1.50 |
|  | *MacsCorr* | 1.25 | 1.69 |
|  | 1400 instances | 1.25 | 8.49 |

Table 8 shows that the 2-Items Reinsertion mutation operator causes the EGGA to be much slower. Said computational cost is closely related to the rearrangement strategy Assemble, incorporated to avoid, as far as possible, becoming stuck in a local optima. Although the computational cost of this strategy is high, it is also too useful, since the properties and characteristics of the addressed problem make the mutation operator by itself incapable of avoiding local optima., mainly in the instances with processing times generated in the ranges $U(1, 100)$ and $U(10, 100)$, where the average times increased from 1.25 to 34.09 and 14.04 seconds, respectively. To review such algorithmic behavior, we analyzed the average generation in which the state-of-the-art GGA and the EGGA find the best solution for each test instance.

Table 9 shows that the GGA becomes quickly trapped in local optima in generation 16 on average, while the EGGA shows a better ability to deal with the landscape characteristics of the $R||C_{max}$ search space, finding its best solutions in generation 362 on average. In this way, Table 9 shows the importance of incorporating the 2-Items Reinsertion mutation operator to the GGA since, although the computational cost is high, it provides to the EGGA a better exploration capability during the search process.

**Table 9.** Comparison of the state-of-the-art GAA and the EGGA based on the generation in which the best solution in population is improved.

|          | Instance         | GGA   | EGGA   |
|----------|------------------|-------|--------|
| $n$      | 100              | 9.27  | 358.34 |
|          | 200              | 8.00  | 369.31 |
|          | 500              | 8.00  | 380.83 |
|          | 1000             | 17.09 | 362.54 |
| $m$      | 10               | 16.35 | 358.46 |
|          | 20               | 13.69 | 359.06 |
|          | 30               | 11.79 | 359.78 |
|          | 40               | 11.05 | 360.96 |
|          | 50               | 10.01 | 362.20 |
| $P_{ij}$ | $U(1,100)$       | 64.80 | 218.56 |
|          | $U(10,100)$      | 8.00  | 305.34 |
|          | $U(100,120)$     | 8.00  | 360.44 |
|          | $U(100,200)$     | 8.00  | 391.96 |
|          | $U(1000,1100)$   | 8.00  | 390.13 |
|          | *JobsCorr*       | 8.00  | 474.82 |
|          | *MacsCorr*       | 8.00  | 392.95 |
|          | 1400 instances   | 16.11 | 362.03 |

From this study, we can conclude that it is still necessary to improve the performance of the EGGA and study its other operators, evaluation function, and stop criteria in order to better explore the search space, since it also becomes stuck in local optima, although not as soon as the original GGA. Additionally, we will focus on analyzing the properties and characteristics of the instances in the sets $U(1,100)$ and $U(10,100)$, where the EGGA stagnates sooner and requires a longer processing time since the rearrangement heuristic is used more times during the solution process of instances with those characteristics.

## 6. Conclusions and Paths of Work

The GGA has become one of the most outstanding metaheuristics for the solution of combinatorial optimization problems related to the partition of a set of items into different subsets. The development of a GGA involves the definition of variation operators adapted to work at the group level. The main goal of this paper was to promote the design of intelligent operators for GGAs as a more suitable way to obtain high-performance GGAs that incorporate knowledge of the problem-domain.

We present a systematic experimental examination to gain insights into the importance of each phase involved in the mutation operator of a GGA designed to solve the Parallel-Machine scheduling problem with unrelated machines and makespan minimization ($R||C_{max}$), analyzing whether different strategies actually contribute to the performance of the operator. The overall procedure of a grouping mutation operator for $R||C_{max}$ comprises: (1) selecting one or more machines; (2) selecting one or more jobs from each of the selected machines; and (3) reinserting the selected jobs in some of the machines. In order to learn something about each of these three algorithmic components, this work covered the analysis of each component in isolation by evaluating distinct strategies to deal with it. In this way, the study covered the evaluation of four state-of-the-art grouping mutation operators, thirty-five operators with different numbers of machines and jobs handled, four machine selection strategies, and two rearrangement heuristics for the reinsertion of the selected jobs. The experimental results suggested that the mutation operator with the best performance: (1) selects two machines, one of the machines with the worst $C_i$ value and one random machine; (2) selects one random job from each of the selected machines; and (3) reinserts the selected jobs in two stages. First, for each job, each machine is checked in an attempt to insert the job in the machine with the lowest $C_i$ value. Second, if the first stage yields the original solution, a rearrangement heuristic is applied to attempt to reduce the processing time of the selected machines by trying to insert one of their jobs into the other machines or to exchange one of their jobs with one job of the other machines.

The knowledge gained from the systematic study was used to design a new grouping mutation operator, called 2-Items Reinsertion. The new operator was incorporated into the state-of-the-art GGA (replacing the original mutation operator) to solve 1400 benchmark instances, showing significant differences with an improvement rate of 52%. These results underline the importance of evaluating the performance of the different components of the GGA operators.

We are aware that the current performance of the Enhanced GGA (EGGA) is still far from reaching the performance of state-of-the-art algorithms for $R||C_{max}$. However, the improvements achieved with the approach proposed in this work are quite promising. Therefore, we believe that with the design and implementation of experimental approaches such as the one presented in this paper we can further improve the performance of EGGA by studying the behavior of other genetic components, such as the population initialization strategy, the selection mechanism, the crossover operator, the replacement mechanism, and the objective function. In this order of ideas, the study of the final performance obtained by the EGGA for the $R||C_{max}$ problem revealed that there still are benchmark instances that show a high degree of difficulty; for these instances, the included strategies in the EGGA do not appear to lead to better solutions. Future work will consist of studying the different components of each operator and technique included in the EGGA, designing a better crossover operator, implementing an efficient reproduction technique, and analyzing the EGGA behavior to understand the impact of each strategy when solving different instances of the $R||C_{max}$ problem. We are also developing a new fitness function that will allow us to discriminate between solutions with the same $C_{max}$ value but with a different exploitation of the machine's processing time. The knowledge gained from the analysis of each component of the grouping mutation operator for the $R||C_{max}$ problem can help us gain a better understanding of the performance of other heuristics for this problem and opens up an interesting range of possibilities for future research on other Parallel-Machine Scheduling variants. It is expected that the study presented in this paper represents a guideline to carry out similar systematic experimental examinations to analyze the components of other GGAs. This knowledge can be used to develop new intelligent operators for solving NP-hard grouping problems.

**Author Contributions:** Conceptualization, O.R.-F., M.Q.-C., E.M.-M. and N.C.-R.; methodology, O.R.-F. and M.Q.-C.; software, O.R.-F.; validation, O.R.-F. and M.Q.-C.; formal analysis, M.Q.-C.; investigation, O.R.-F. and M.Q.-C.; resources, O.R.-F.; writing—original draft preparation, O.R.-F.; writing—review and editing, O.R.-F., M.Q.-C., E.M.-M. and N.C.-R.; visualization, O.R.-F. and M.Q.-C.; supervision, E.M.-M. and M.Q.-C.; project administration, M.Q.-C. All authors have read and agreed to the published version of the manuscript.

**Funding:** This research received no external funding.

**Conflicts of Interest:** The authors declare no conflict of interest. The funders had no role in the design of the study; in the collection, analyses, or interpretation of data; in the writing of the manuscript, or in the decision to publish the results.

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
