# Peer review of "An Experimental Study of Grouping Mutation Operators for the Unrelated Parallel-Machine Scheduling Problem"

_mca, doi:10.3390/mca28010006_

Round 1
Reviewer 1 Report
Dear respected authors,
1. The main aims and the methods used in the study have been reflected well in the Abstract section.
2. It is suggested to add the Nomenclature list before the Introduction section for ease of readiness by potential readers.
3. The combinatorial optimization problem should be defined in the Introduction section.
4. After defining an Acronym/Abbreviation, it should be used instead of the whole phrase everywhere in the text and there is no need to define it again and again. For instance, GGA has been defined in line 75, and again in lines 92 and 99, which are unnecessary. The other Abbreviations/Acronyms used in the text must be checked considering this point of view.
5. The sentence “This problem family has received much recognition due to its numerous real-world applications.” in line 37 needs to be supported by 2 or 3 recent highly cited references.
6. The sentence “Among the most recent solution proposals are exact algorithms like the Branch and bound” in line 38 is not a complete sentence. It should be rewritten.
7. In lines 63 to 65, many articles have been cited that are not only unnecessary, but also many of them are too old, and related to more than decades ago. It is recommended to cite 2 or 3 references for each fact or sentence in this part of the article, and try to find the most related ones.
8. The content of Section 2, needs to be more supported by recent references.
9. In section 3, just four types of mutation operators have been used. As we know, there are various methods for mutation in the literature. Therefore, the question of “Why these four operators” must be answered and explained in the text.
10. Conclusion section should be rearranged. Any results or comparisons should be mentioned before this section. Specifically, Table 8 and its explanation and discussions should be added to the previous section.
Author Response
Response to Reviewer 1 Comments
Point 1: The main aims and the methods used in the study have been reflected well in the Abstract section.
Response 1: Thanks for the observation.
Point 2: It is suggested to add the Nomenclature list before the Introduction section for ease of readiness by potential readers.
Response 2: Thanks for the observation. We improved the quality of the work by incorporating the nomenclature before the Introduction (lines 17-66).
Point 3: The combinatorial optimization problem should be defined in the Introduction section.
Response 3: Thanks for the observation. We enhanced the information in the Introduction by including the definition of a COP (Section 1, lines 70-74).
Point 4: After defining an Acronym/Abbreviation, it should be used instead of the whole phrase everywhere in the text and there is no need to define it again and again. For instance, GGA has been defined in line 75, and again in lines 92 and 99, which are unnecessary. The other Abbreviations/Acronyms used in the text must be checked considering this point of view.
Response 4: Thanks for the observation. We edited the work to keep only the first definition for each Acronym/Abbreviation only, and we use the Acronym/Abbreviation in rest of the text (lines 1, 68, 80, 91, 120, 121, 124, 197, 232).
Point 5: The sentence “This problem family has received much recognition due to its numerous real-world applications.” in line 37 needs to be supported by 2 or 3 recent highly cited references.
Response 5: Thanks for the observation. We updated this text (lines 92 y 93) by adding three highly cited references to support the sentence “This problem family has received much recognition due to its numerous real-world applications”. We used the following references:
- Fanjul-Peyro, L. (2020). Models and an exact method for the unrelated parallel machine scheduling problem with setups and resources. Expert Systems with Applications: X, 5, 100022.
- Bitar, A., Dauzère-Pérès, S., & Yugma, C. (2021). Unrelated parallel machine scheduling with new criteria: Complexity and models. Computers & Operations Research, 132, 105291.
- Moser, M., Musliu, N., Schaerf, A., & Winter, F. (2022). Exact and metaheuristic approaches for unrelated parallel machine scheduling. Journal of Scheduling, 25(5), 507-534.
Point 6: The sentence “Among the most recent solution proposals are exact algorithms like the Branch and bound” in line 38 is not a complete sentence. It should be rewritten.
Response 6: Thanks for the observation. We changed the sentence “Although a large number of mathematical models have been proposed, the exact approaches can solve in reasonable time only small instances." (lines 93 and 94).
Point 7: In lines 63 to 65, many articles have been cited that are not only unnecessary, but also many of them are too old, and related to more than decades ago. It is recommended to cite 2 or 3 references for each fact or sentence in this part of the article, and try to find the most related ones.
Response 7: Thanks for the observation. We reduced the number of references, discarding the oldest and least related ones (lines 117-120).
Point 8: The content of Section 2, needs to be more supported by recent references.
Response 8: Thanks for the observation. We incorporated three recent references to support Section 2 and show that GGA is still widely used to solve grouping problems (lines 155-158).
- Ramos-Figueroa, O.; Quiroz-Castellanos, M.; Mezura-Montes, E.; Schütze, O. Metaheuristics to solve grouping problems: A review and a case study. Swarm and Evolutionary Computation 2020, 53, 100643.
- Carmona-Arroyo, G.; Quiroz-Castellanos, M.; Mezura-Montes, E. Variable Decomposition for Large-Scale Constrained Optimization Problems Using a Grouping Genetic Algorithm. Mathematical and Computational Applications 2022, 27, 23.
- Alharbe, N.; Aljohani, A.; Rakrouki, M.A. A Fuzzy Grouping Genetic Algorithm for Solving a Real-World Virtual Machine Placement Problem in a Healthcare-Cloud. Algorithms 2022, 15, 128.
Point 9: In section 3, just four types of mutation operators have been used. As we know, there are various methods for mutation in the literature. Therefore, the question of “Why these four operators” must be answered and explained in the text.
Response 9: Thanks for the observation. We improved the paper by highlighting that this work only contemplates the best state-of-the-art mutation operators that apply for the R||Cmax problem, discarding the infeasible ones and those which have not shown outstanding performance
(lines 387-393). Furthermore, we return to this idea in Section 4.1 to recall why we use only these four operators and remark that there are others (lines 456-460).
Point 10: Conclusion section should be rearranged. Any results or comparisons should be mentioned before this section. Specifically, Table 8 and its explanation and discussions should be added to the previous section.
Response 10: Thanks for the observation. We enhanced the work by relocating the tables to place them in the sections were they are defined and described (Table 9, page 27). It is important to note that the Table 8 now is Table 9 since a new table was incorporated.

Reviewer 2 Report
The authors present a study of different grouping mutation operators for the Unrelated Parallel-Machine Scheduling Problem.
I would mention my observations to improve their work:
i) Authors cannot assert that NVST-IG+ is the best solution for the problem in all cases (Lines 68-70).
ii) Authors cannot assert that "standard" GA and PSO are worse than GGA; "standard" operators are confused (Lines 74-76).
iii) Please present the GGA in 2 by name or nomenclature in the entire manuscript (Lines 83-89).
iv) Include the citation of the Min Heuristic (Line 154).
v) RPD is not previously presented (Line 282)
vi) It is necessary to include the Equations and Pseudocode from all the indicators, selection strategies, crossover, and mutation operators in the entire manuscript. Text descriptions are hard to read, and it would improve the manuscript.
vii) The experimental settings and parameters are not precise. Please include them in a Table format.
Author Response
Response to Reviewer 2 Comments
Point 1: Authors cannot assert that NVST-IG+ is the best solution for the problem in all cases (Lines 68-70).
Response 1: Thanks for the observation. We updated the work by remplacing the sentence "considered the best solution method designed for the problem of interest" by the sentence "considered one of the best solution method designed for the problem of interest so far" (line 123-125).
Point 2: Authors cannot assert that "standard" GA and PSO are worse than GGA; "standard" operators are confused (Lines 74-76).
Response 2: Thanks for the observation. We modified the text by specifying that the experimental results suggest that GGA has a better performance than a GA with an extended permutation solution encoding and a PSO with a machine-based representation scheme for the test instances employed (line 129-132).
Point 3: Please present the GGA in 2 by name or nomenclature in the entire manuscript (Lines 83-89).
Response 3: Thanks for the observation. We unifyed the text by incorporating the nomenclature before the Introduction (lines 17-66). Furthermore, we defined each Acronym/Abbreviation only the fist time and we used the Acronym/Abbreviation in rest of the text.
Point 4: Include the citation of the Min Heuristic (Line 154).
Response 4: Thanks for the observation. We enhanced this sentence by adding the reference of the Min() heuristic (lines 209 and 210).
- Ibarra, O.H.; Kim, C.E. Heuristic algorithms for scheduling independent tasks on nonidentical processors. Journal of the ACM (JACM) 1977, 24, 280–289.
Point 5: RPD is not previously presented (Line 282).
Response 5: Thanks for the observation. We edited the text by replacing the sentence "lower RPD" by the sentence "lower error rate" to avoid confusions and preserve the structure of the work (line 360). In addition, we also defined $RPD$ in the new Nomenclature Section, included before the introduction, to make the text easier to understand (line 52).
Point 6: It is necessary to include the Equations and Pseudocode from all the indicators, selection strategies, crossover, and mutation operators in the entire manuscript. Text descriptions are hard to read, and it would improve the manuscript.
Response 6: Thanks for the observation. We improved the work by incorporating eight algorithms (Sections 2 and 4, Algorithms 1-8) with their respective descriptions to facilitate the understanding of all the Equations and Pseudocode used.
Point 7: The experimental settings and parameters are not precise. Please include them in a Table forma
Response 7: Thanks for the observation. We improved the parameter configuration description by incorporating a table to facilitate its location within the text (Table 5, page 24).

Round 2
Reviewer 1 Report
Dear respected authors,
The methodology, procedures, and steps of the research have been done logically. Thank the authors for modifying and answering all the comments and suggestions, precisely. According to the reviewer's point of view, the manuscript is worth being published in the respected journal.